



# Global inorganic nitrate production mechanisms:
# Comparison of a global model with nitrate isotope
# observations
Becky Alexander[1], Tomás Sherwen[2,3], Christopher D. Holmes[4], Jenny A. Fisher[5], Qianjie
Chen[1,6], Mat J. Evans[2,3], Prasad Kasibhatla[7]
[1]Department of Atmospheric Sciences, University of Washington, Seattle, WA 98195, USA
[2]Wolfson Atmospheric Chemistry Laboratories, Department of Chemistry, University of York, York YO10 5DD, UK
[3]National Center for Atmospheric Science, University of York, York YO10 5DD, UK
[4]Department of Earth, Ocean and Atmospheric Science, Florida State University, Tallahassee, FL 32306, USA
[5]Centre for Atmospheric Chemistry, University of Wollongong, Wollongong, New South Wales 2522, Australia
[6]Now at Department of Chemistry, University of Michigan, Ann Arbor, MI 48109, USA
[7]Nicholas School of the Environment, Duke University, Durham, NC 27708, USA
Correspondence to: Becky Alexander (beckya@uw.edu)
**Abstract.** The formation of inorganic nitrate is the main sink for nitrogen oxides ($NO_x = NO + NO_2$). Due to the
importance of $NO_x$ for the formation of tropospheric oxidants such as the hydroxyl radical (OH) and ozone,
understanding the mechanisms and rates of nitrate formation is paramount for our ability to predict the atmospheric
lifetimes of most reduced trace gases in the atmosphere. The oxygen isotopic composition of nitrate ($\Delta^{17}O(nitrate)$) is
determined by the relative importance of $NO_x$ sinks, and thus can provide an observational constraint for $NO_x$
chemistry. Until recently, the ability to utilize $\Delta^{17}O(nitrate)$ observations for this purpose was hindered by our lack
of knowledge about the oxygen isotopic composition of ozone ($\Delta^{17}O(O_3)$). Recent and spatially widespread
observations of $\Delta^{17}O(O_3)$ have greatly reduced this uncertainty, and allow for an updated comparison of modeled and





observed $\Delta^{17}$O(nitrate) and a reassessment of modeled nitrate formation pathways.  Model updates based on recent
laboratory studies of heterogeneous reactions renders dinitrogen pentoxide ($N_2O_5$) hydrolysis as important as $NO_2$ +
OH (both 41%) for global inorganic nitrate production near the surface.  All other nitrate production mechanisms
represent less than 6% of global nitrate production near the surface, but can be dominant locally.  Updated reaction
rates for aerosol uptake of $NO_2$ result in significant reduction of nitrate and nitrous acid ( HONO) formed through this
pathway in the model, and render $NO_2$ hydrolysis a negligible pathway for nitrate formation globally. Although
photolysis of aerosol nitrate may have implications for $NO_x$, HONO and oxidant abundances, it does not significantly
impact the relative importance of nitrate formation pathways.    Modeled $\Delta^{17}$O(nitrate) (28.6 ± 4.5‰) compares well
with the average of a global compilation of observations (27.6 ± 5.0‰), giving confidence in the model's
representation of the relative importance of ozone versus $HO_x$ (= OH + $HO_2$ + $RO_2$) in $NO_x$ cycling and nitrate
formation.
**1.  Introduction**
Nitrogen oxides ($NO_x$ = NO + $NO_2$) are a critical ingredient for the formation of tropospheric ozone ($O_3$).
Tropospheric ozone is a greenhouse gas, is a major precursor for the hydroxyl radical (OH), and is considered an air
pollutant due to its negative impacts on human health. The atmospheric lifetime of $NO_x$ is determined by its oxidation
to inorganic and organic nitrate. The formation of inorganic nitrate ($HNO_3$(g) and particulate $NO_3^-$) is the dominant
sink for $NO_x$ globally, while formation of organic nitrate may be significant in rural and remote continental locations
(Browne and Cohen, 2014). Organic nitrate as a sink for $NO_x$ may be becoming more important in regions in North
America and Europe where $NO_x$ emissions have declined (Zare et al., 2018).  Uncertainties in the rate of oxidation of
$NO_x$ to nitrate has been shown to represent a significant source of uncertainty for ozone and OH formation in models
(e.g., Newsome and Evans (2017)) , with implications for our understanding of the atmospheric lifetime of species
such as methane, whose main sink is reaction with OH.
$NO_x$ is emitted to the atmosphere primarily as NO by fossil fuel and biomass/biofuel burning, soil microbes, and
lightning.  Anthropogenic sources from fossil fuel and biofuel burning and from the application of fertilizers to soil
for agriculture currently dominate $NO_x$ sources to the atmosphere (Jaeglé et al., 2005).  After emission, NO is rapidly



oxidized to $NO_2$ by ozone ($O_3$), peroxy ($HO_2$) and hydroperoxy radicals ($RO_2$), and halogen oxides (e.g., BrO).  During
the daytime, $NO_2$ is rapidly photolyzed to $NO + O$ at wavelengths ($\lambda$) < 424 nm.  $NO_x$ cycling between NO and $NO_2$
proceeds several orders of magnitude faster than oxidation of $NO_x$ to nitrate during the daytime (Michalski et al.,

4   2003).

Formation of inorganic nitrate is dominated by oxidation of $NO_2$ by OH during the day and by the hydrolysis of
dinitrogen pentoxide ($N_2O_5$) at night (Alexander et al., 2009).  Recent implementation of reactive halogen chemistry
in models of tropospheric chemistry show that formation of nitrate from the hydrolysis of halogen nitrates ($XNO_3$,
where X = Br, Cl, or I) is also a sink for $NO_x$ with implications for tropospheric ozone, OH, reactive halogens, and
aerosol formation (Schmidt et al., 2016;Sherwen et al., 2016;Saiz-Lopez et al., 2012;Long et al., 2014;Parrella et al.,
2012;von Glasow and Crutzen, 2004).  Other inorganic nitrate formation pathways include hydrogen-abstraction of
hydrocarbons by the nitrate radical ($NO_3$), heterogeneous reaction of $N_2O_5$ with particulate chloride ($Cl^-$),
heterogeneous uptake of $NO_2$ and $NO_3$, direct oxidation of NO to $HNO_3$ by $HO_2$, and hydrolysis of organic nitrate.
Inorganic nitrate partitions between the gas ($HNO_3(g)$) and particle ($NO_3^-$) phases, with its relative partitioning
dependent upon aerosol abundance, aerosol liquid water content, aerosol chemical composition, and temperature.
Inorganic nitrate is lost from the atmosphere through wet or dry deposition to the Earth's surface with a global lifetime
against deposition on the order of 3-4 days (Alexander et al., 2009).
Formation of inorganic nitrate was thought to be a permanent sink for $NO_x$ in the troposphere due to the slow
photolysis of nitrate compared to deposition.  However, laboratory and field studies have shown that $NO_3^-$ adsorbed
on surfaces is photolyzed at rates much higher than $HNO_3(g)$ (Ye et al., 2016).  The photolysis of $NO_3^-$ in snow grains
on ice sheets has a profound impact on the oxidizing capacity of the polar atmosphere (Domine and Shepson, 2002).
More recently, observations of $NO_x$ and nitrous acid (HONO) provide evidence of photolysis of aerosol $NO_3^-$ in the
marine (Reed et al., 2017;Ye et al., 2016) and continental (Ye et al., 2018;Chen et al., 2019) boundary layer, with
implications for ozone and OH (Kasibhatla et al., 2018).
Organic nitrates form during reaction of $NO_x$ and $NO_3$ with biogenic volatile organic compounds (BVOCs) and their
oxidation products (organic peroxy radicals, $RO_2$) (Browne and Cohen, 2014;Liang et al., 1998).  Products of these





reactions include peroxy nitrates ($RO_2NO_2$) and alkyl and multifunctional nitrates ($RONO_2$) (O'Brien et al., 1995).
Peroxy nitrates are thermally unstable and decompose back to $NO_x$ on the order of minutes to days at warm
temperatures. Decomposition of longer-lived peroxy nitrates such as peroxyacetyl nitrate (PAN) can provide a source
of $NO_x$ to remote environments (Singh et al., 1992). The fate of $RONO_2$ is uncertain. First-generation $RONO_2$ is
oxidized to form second-generation $RONO_2$ species with a lifetime of about a week for the first-generation species
with $\geq$ 4 carbon atoms, and up to several weeks for species with fewer carbon atoms (e.g., days to weeks for methyl
nitrate) (Fisher et al., 2018). Subsequent photolysis and oxidation of second-generation $RONO_2$ species can lead to
the recycling of $NO_x$ (Müller et al., 2014), although recycling efficiencies are highly uncertain (Horowitz et al.,
2007;Paulot et al., 2009). $RONO_2$ can also partition to the particle phase ($pRONO_2$) contributing to organic aerosol
formation (Xu et al., 2015). $pRONO_2$ is removed from the atmosphere by deposition to the surface, or through
hydrolysis to form inorganic nitrate and alcohols (Rindelaub et al., 2015;Jacobs et al., 2014).
The oxygen isotopic composition ($\Delta^{17}O = \delta^{17}O - 0.52 \times \delta^{18}O$) of nitrate is determined by the relative importance of
oxidants leading to nitrate formation from the oxidation of $NO_x$ (Michalski et al., 2003). Observations of the oxygen
isotopic composition of nitrate ($\Delta^{17}O(nitrate)$) have been used to quantify the relative importance of different nitrate
formation pathways and to assess model representation of the chemistry of nitrate formation in the present day
(Alexander et al., 2009;Michalski et al., 2003;Costa et al., 2011;Ishino et al., 2017a;Morin et al., 2009;Morin et al.,
2008;Savarino et al., 2007;Kunasek et al., 2008;Savarino et al., 2013;McCabe et al., 2007;Morin et al., 2007;Hastings
et al., 2003;Kaiser et al., 2007;Brothers et al., 2008;Ewing et al., 2007) and in the past from nitrate archived in ice
cores (Sofen et al., 2014;Alexander et al., 2004;Geng et al., 2014;Geng et al., 2017). Ozone-influenced reactions in
$NO_x$ oxidation lead to high $\Delta^{17}O(nitrate)$ values while $HO_x$-influenced reactions lead to $\Delta^{17}O(nitrate)$ near zero.
Oxidation by XO (where X = Br, Cl, or I) leads to $\Delta^{17}O(nitrate)$ values similar to reactions with ozone because the
oxygen atom in XO is derived from the reaction $X + O_3$. Therefore, $\Delta^{17}O(nitrate)$ is determined by the relative
importance of $O_3 + XO$ versus $HO_x$ (= $OH + HO_2 + RO_2$) in both $NO_x$ cycling and oxidation to nitrate. Due to rapid
$NO_x$-cycling during the daytime, $NO_x$ achieves isotopic equilibrium and its $\Delta^{17}O$ value ($\Delta^{17}O(NO_x)$) is solely
determined by the relative abundance of ($O_3 + XO$) to ($HO_2 + RO_2$).



The $\Delta^{17}O$ value of $HO_x$ ($\Delta^{17}O(HO_x)$) is near zero due to isotopic exchange of OH with water vapor (Dubey et al.,
1997).    The $\Delta^{17}O$ value of ozone ($\Delta^{17}O(O_3)$) was until recently not well known due to uncertainties arising from
sampling artifacts in the earlier measurements (Johnston and Thiemens, 1997;Krankowsky et al., 1995) and has been
the largest source of uncertainty in quantification of nitrate formation pathways using observations of $\Delta^{17}O(nitrate)$
(Alexander et al., 2009). Previous modeling studies showed good agreement with observations of $\Delta^{17}O(nitrate)$ when
assuming $\Delta^{17}O(O_3)$ = 35‰ (Alexander et al., 2009;Michalski et al., 2003).    Recently, much more extensive
observations of $\Delta^{17}O(O_3)$ using a new technique (Vicars et al., 2012) show $\Delta^{17}O(O_3)$ = 26 ± 1‰ around the globe
(Vicars et al., 2012;Ishino et al., 2017b;Vicars and Savarino, 2014), and suggest that previous modeling studies are
biased low in $\Delta^{17}O(nitrate)$ (e.g., Alexander et al. (2009)), which would occur if the model underestimated the relative
role of ozone in $NO_x$ chemistry.    Reduction in uncertainty in the value of $\Delta^{17}O(O_3)$ enables improved interpretation of
$\Delta^{17}O(nitrate)$ as an observational constraint for the relative importance of nitrate formation pathways in the
atmosphere.    Here, we examine the relative contribution of each nitrate formation pathway in a global chemical
transport model and compare the model with observations of $\Delta^{17}O(nitrate)$ from around the world.
**2.  Methods**
We use the GEOS-Chem global chemical transport model version 12.0.0 driven by assimilated meteorology from the
MERRA-2 reanalysis product with a native resolution of 0.5° x 0.625° and 72 vertical levels from the surface up to
the 0.01 hPa pressure level.  For computational expediency, the horizontal and vertical resolution were downgraded
to 4° x 5° and 47 vertical levels.   GEOS-Chem was originally described in Bey et al. (2001) and includes coupled
$HO_x$-$NO_x$-VOC-ozone-halogen-aerosol tropospheric chemistry as described in Sherwen et al. (2016) and Sherwen et
al. (2017) and organic nitrate chemistry as described in Fisher et al. (2016). Aerosols interact with gas-phase chemistry
through the effect of aerosol extinction on photolysis rates (Martin et al., 2003) and heterogeneous chemistry (Jacob,
2000).  The model calculates deposition for both gas species and aerosols (Liu et al., 2001;Zhang et al., 2001;Wang
et al., 1998).
Global anthropogenic emissions, including $NO_x$, are from the Community Emissions Data System (CEDS) inventory
from 1950 – 2014 C.E. (Hoesly et al., 2018a).  The CEDS global emissions inventory is overwritten by  regional




anthropogenic emissions inventories in the U.S. (EPA/NE11), Canada (CAC), Europe (EMEP), and Asia (MIX (Li et
al., 2017)). Global shipping emissions are from the International Comprehensive Ocean-Atmosphere Data Set
(ICOADS), which was implemented into GEOS-Chem as described in Lee et al. (2011). $NO_x$ emissions from ships
are processed using the PARANOX module described in Vinken et al. (2011) and Holmes et al. (2014) to account for
non-linear, in-plume ozone and $HNO_3$ production. Lightning $NO_x$ emissions match the OTD/LIS satellite
climatological observations of lightning flashes as described by Murray et al. (2012). Emissions from open fires are
from the Global Fire Emissions Database (GFED4.1). Biogenic soil $NO_x$ emissions are described in Hudman et al.
(2012). Aircraft emissions are from the Aviation Emissions Inventory Code (AEIC) (Stettler et al., 2011).
Chemical processes leading to nitrate formation in GEOS-Chem have expanded since the previous work of Alexander
et al. (2009). Figure 1 summarizes the formation of inorganic nitrate in the current model. In the model, NO is
oxidized by $O_3$, $HO_2$, $RO_2$ and halogen oxides (XO = BrO, ClO, IO, and OIO) to form $NO_2$. $NO_2$ can form $HNO_3$
directly from its reaction with OH and $HO_2$ and through hydrolysis on aerosol surfaces. $NO_2$ can react with XO to
form halogen nitrates ($BrNO_3$, $ClNO_3$, and $INO_3$), which can then form $HNO_3$ upon hydrolysis (as described in
Sherwen et al. (2016)). $NO_2$ can also react with $O_3$ to form $NO_3$, which can then react with $NO_2$, hydrocarbons (HC),
and the biogenic VOCs monoterpenes (MTN) and isoprene (ISOP). Reaction of $NO_3$ with $NO_2$ forms $N_2O_5$, which
can subsequently hydrolyze or react with $Cl^-$ in aerosol to form $HNO_3$. Reaction of $NO_3$ with HC forms $HNO_3$ via
hydrogen abstraction. Reactions of $NO_3$ are only important at night due to its short lifetime against photolysis.
Formation of organic nitrate ($RONO_2$) was recently updated in the model as described in Fisher et al. (2016). Reaction
of $NO_3$ with MTN and ISOP can form $RONO_2$. $RONO_2$ also forms from the reaction of NO with $RO_2$ derived from
OH oxidation of BVOCs. $RONO_2$ hydrolyzes to form $HNO_3$ on a timescale of 1 hour. Inorganic nitrate partitions
between the gas ($HNO_3(g)$) and particle ($NO_3^-$) phase according to local thermodynamic equilibrium as calculated in
the ISORROPIA-II aerosol thermodynamic module (Fountoukis and Nenes, 2007). $HNO_3(g)$ and $NO_3^-$ are mainly
lost from the atmosphere via wet and dry deposition to the surface.
In the "standard" model, hydrolysis of $N_2O_5$, $NO_3$ ($\gamma_{NO3} = 1 \times 10^{-3}$), and $NO_2$ ($\gamma_{NO2} = 1 \times 10^{-4}$) occur on aerosol surfaces
only. Uptake and hydrolysis of $N_2O_5$ on aerosol surfaces depends on the chemical composition of aerosols,
temperature, and humidity as described in Evans and Jacob (2005). Recently, Holmes et al. (2019) updated the



reaction probabilities of the $NO_2$ and $NO_3$ heterogeneous reactions in the model to depend on aerosol chemical
composition and relative humidity. Holmes et al. (2019) also updated the $N_2O_5$ reaction probability to additionally
depend on the $H_2O$ and $NO_3^-$ concentrations in aerosol (Bertram and Thornton, 2009). In addition to these updates
for hydrolysis on aerosol, Holmes et al. (2019) included the uptake and hydrolysis of $N_2O_5$, $NO_2$, and $NO_3$ in cloud
water and ice limited by cloud entrainment rates. We incorporate these updates from Holmes et al. (2019) into the
"cloud chemistry" model to examine the impacts on global nitrate production mechanisms. We consider the "cloud
chemistry" model as state-of-the science, and as such we focus on the results of this particular simulation. Additional
model sensitivity studies are also performed and examined relative to the "standard" model simulation. These
additional sensitivity simulations are described in Section 4.
$\Delta^{17}O$(nitrate) is calculated in the model using monthly-mean, local chemical production rates, rather than by treating
different isotopic combinations of nitrate as separate tracers that can be transported in the model. Alexander et al.
(2009) transported four nitrate tracers, one each for nitrate production by $NO_2$+OH, $N_2O_5$ hydrolysis, $NO_3$+HC, and
nitrate originating from its formation in the stratosphere. Since $\Delta^{17}O(NO_x)$ was not transported in the Alexander et al.
(2009) model, it was calculated using local production rates, so effectively only one-third of the $\Delta^{17}O$(nitrate) was
transported in Alexander et al. (2009). Accurately accounting for transport of $\Delta^{17}O$(nitrate) in the model would require
transporting all individual isotopic combinations of the primary reactant (NO), the final product (nitrate), and each
reaction intermediate (e.g., $N_2O_5$), which we do not do here due to the large computational costs. Thus, the model
results shown here represent $\Delta^{17}O$(nitrate) from local $NO_x$ cycling and nitrate production. This may lead to model
biases, particularly in remote regions such as polar-regions in winter-time when most nitrate is likely transported from
lower latitudes or the stratosphere. This should make little difference in polluted regions where most nitrate is formed
locally. This approach will however reflect the full range of possible modeled $\Delta^{17}O$(nitrate) values, which can then
be compared to the range of observed $\Delta^{17}O$(nitrate) values.
The $\Delta^{17}O$(nitrate) value of nitrate produced from each production pathway is calculated as shown in Table 1. The
value of $A$ in Table 1 represents the relative importance of the oxidation pathways of NO to $NO_2$ where the oxygen
atom transferred comes from ozone (NO + $O_3$ and NO + XO):



$$A = \frac{k_{O_3+NO}[O_3] + k_{XO+NO}[XO]}{k_{O_3+NO}[O_3] + k_{XO+NO}[XO] + k_{HO_2+NO}[HO_2] + k_{RO_2+NO}[RO_2]} \qquad \text{(E1)}$$
In E1, $k$ represents the local reaction rate constant for each of the four reactions, XO = BrO, ClO, IO, and OIO, and
we assume $\Delta^{17}O(XO)$ is equal to the $\Delta^{17}O$ value of the terminal oxygen atoms of ozone, as described in more detail
below. This effectively assumes that the other oxidation pathways (NO + HO$_2$ and NO + RO$_2$) yield $\Delta^{17}O(NO_x) =$
0‰. Although HO$_2$ may have a small $^{17}O$ enrichment on the order of 1-2‰ (Savarino and Thiemens, 1999b), the
assumption that this pathway yields $\Delta^{17}O(NO_x) = 0$‰ simplifies the calculation and leads to negligible differences in
calculated $\Delta^{17}O(nitrate)$ (Michalski et al., 2003). This approach assumes that NO$_x$ cycling is in photochemical steady-
state, which only occurs during the daytime. $A$ is calculated in the model as the 24-hour average NO$_2$ production rate,
rather than the daytime average only. As was shown in Alexander et al. (2009), rapid daytime NO$_x$ cycling dominates
the calculated 24-hour averaged $A$ value, leading to negligible differences in calculated $\Delta^{17}O(nitrate)$ for 24-hour
averaged values versus daytime averaged values.
NO$_x$ formed during the day will retain its daytime $\Delta^{17}O(NO_x)$ signature throughout the night due to lack of NO$_2$
photolysis (Morin et al., 2011), suggesting similar $A$ values for the nighttime reactions (R2, R4, R5, R8, and R10 in
Table 1). However, NO emitted at night will retain its originally emitted isotopic signature ($\Delta^{17}O(NO) = 0$‰) due to
lack of NO$_x$ cycling under dark conditions. Any NO emitted at night and oxidized to NO$_2$ before sunrise will result
in $\Delta^{17}O(NO_2)$ equal to one-half of the $\Delta^{17}O$ value of the oxidant, since only one of the two oxygen atoms of NO$_2$ will
originate from the oxidant. Since HO$_x$ abundance is low at night, ozone will be the dominant oxidant. Thus, NO both
emitted and oxidized to NO$_2$ at night will lead to $A_{night} = 0.5$ (half of the O atoms of NO$_2$ originate from O$_3$). Since
the atmospheric lifetime of NO$_x$ against nighttime oxidation to nitrate (R2+R4+R5) is typically greater than 24 hours
(Figure S1), most nitrate formed during the nighttime will form from NO$_x$ that reached photochemical equilibrium
during the previous day. Thus, we use values of $A$ calculated as the 24-hour average NO$_2$ production rate for
calculating the $\Delta^{17}O(nitrate)$ value of all nitrate production pathways, including those that can occur at night. This is
consistent with a box modeling study that explicitly calculated the diurnal variability of $\Delta^{17}O(NO_x)$ and $\Delta^{17}O(nitrate)$
suggesting similar (within 5%) values for $\Delta^{17}O(nitrate)$ when assuming the NO$_x$ reached photochemical steady-state
versus explicit calculation of diurnal variability of $\Delta^{17}O(NO_x)$ and $\Delta^{17}O(nitrate)$ (Morin et al., 2011). Using 24-hour
averaged $A$ values may lead to an overestimate of $\Delta^{17}O(nitrate)$ in locations with more rapid nighttime nitrate



formation rates such as in China and India (Figure S1). However, even in these locations the lifetime of $NO_x$ against
nighttime oxidation is greater than 12 hours, suggesting that over half of nitrate formation at night occurs from the
oxidation of $NO_x$ that reached photochemical equilibrium during the daytime. When comparing modeled $\Delta^{17}O(nitrate)$
with observations, we add error bars to model values in these locations (Beijing and Mt. Lulin, Taiwan) that reflect
the range of possible $A$ values for nighttime nitrate formation, with the high end ($A_{high}$) reflecting 24-hour average $A$
values and the low end assuming that half of nitrate formation occurs from oxidation of $NO_x$ that reached
photochemical equilibrium during the daytime ($A_{low} = 0.5A + 0.5A_{night} = 0.5A + 0.25$).
$\Delta^{17}O(nitrate)$ for total nitrate is calculated in the model according to:
$$\Delta^{17}O(nitrate) = \sum_{R=R1}^{R10} f_R \Delta^{17}O(nitrate)_R \qquad\qquad\qquad (E2)$$
where $f_R$ represents the fractional importance of each nitrate production pathway (R1-R10 in Table 1) relative to total
nitrate production, and $\Delta^{17}O(nitrate)_R$ is the $\Delta^{17}O(nitrate)$ value for each reaction as described in Table 1. To calculate
$\Delta^{17}O(nitrate)$, we assume that the mean $\Delta^{17}O$ value of the ozone molecule ($\Delta^{17}O(O_3)$) is equal to 26‰ based on recent
observations (Vicars et al., 2012;Ishino et al., 2017b;Vicars and Savarino, 2014). Since the $^{17}O$ enrichment in $O_3$ is
contained entirely in its terminal oxygen atoms (Vicars et al., 2012;Berhanu et al., 2012;Bhattacharya et al.,
2008;Savarino et al., 2008;Michalski and Bhattacharya, 2009;Bhattacharya et al., 2014), and it is the terminal oxygen
atom that is transferred to the oxidation product during chemical reactions (Savarino et al., 2008;Berhanu et al., 2012),
the $\Delta^{17}O$ value of the oxygen atom transferred from ozone to the product is 50% larger than the bulk $\Delta^{17}O(O_3)$ value.
Thus, we assume that the $\Delta^{17}O$ value of the oxygen atom transferred from $O_3$ ($\Delta^{17}O(O_3^*)$) = 1.5 x $\Delta^{17}O(O_3)$ = 39‰, as
in previous work (e.g., (Morin et al., 2011)), where $\Delta^{17}O(O_3^*)$ represents the $\Delta^{17}O$ value of the terminal oxygen atoms
in ozone.
**3. Results and Discussion**
Figure 1 shows the relative importance of the different oxidation pathways of NO to $NO_2$ and nitrate formation below
1 km altitude in the model for the "cloud chemistry" simulation, with equivalent values for the "standard" simulation





shown in parentheses. We focus on model results near the surface because these can be compared to observations;
currently only surface observations of $\Delta^{17}O$(nitrate) are available. The dominant oxidant of NO to $NO_2$ is $O_3$ (84-
85%). Much of the remaining oxidation occurs due to the reaction with peroxy radicals ($HO_2$ and $RO_2$). Oxidation of
NO to $NO_2$ by XO is minor (1%) and occurs over the oceans because the main source of tropospheric reactive halogens
is from sea salt aerosol and sea water (Chen et al., 2017;Sherwen et al., 2016;Wang et al., 2018) (Figure 2).
For both the "cloud chemistry" and "standard" simulations, the two most important nitrate formation pathways are
$NO_2$ + OH (41-42%) and $N_2O_5$ hydrolysis (28-41%) , the latter of which is dominant over the mid- to high-northern
continental latitudes during winter where both $NO_x$ emissions and aerosol abundances are relatively large (Figures 1
and 3). The "cloud chemistry" simulation results in an equal importance of nitrate formation via $NO_2$ + OH and $N_2O_5$
hydrolysis (both 41%) due to increases in the rate of $N_2O_5$ uptake in clouds and decreases in the importance of $NO_2$
hydrolysis, which can compete with $N_2O_5$ formation at night. In the "standard" model, $NO_2$ hydrolysis represents an
important nitrate production mechanism (12%), but it is negligible in the "cloud chemistry" simulation due to the
reduction in the reaction probability (from $\gamma_{NO2} = 10^{-4}$ to $\gamma_{NO2} = 10^{-4}$ to $10^{-8}$) in the model, which is supported by
laboratory studies (Burkholder et al., 2015;Crowley et al., 2010;Tan et al., 2016). The formation of $HNO_3$ from the
hydrolysis of $RONO_2$ formed from both daytime (NO + $RO_2$) and nighttime ($NO_3$ + MTN/ISOP) reactions represents
6% of total, global nitrate formation (Figure 1) and is dominant over Amazonia (Figure 3). $RONO_2$ hydrolysis
represents up to 20% of inorganic nitrate formation in the southeast U.S. (Figure 3). This is similar to Fisher et al.
(2016) who estimated that formation of $RONO_2$ accounts for up to 20% of $NO_x$ loss in this region during summer,
with $RONO_2$ hydrolysis representing 60% of $RONO_2$ loss. Globally, the formation of inorganic nitrate from the
hydrolysis of $RONO_2$ is dominated by $RONO_2$ formation from the daytime reactions (3-6%), while the formation of
$RONO_2$ from nighttime reactions represents up to 3%. The relative importance of nighttime and daytime $RONO_2$
formation is expressed as a range because precursors to $RONO_2$ that formed from monoterpenes can form from both
daytime and nighttime reactions, and these precursors are not separately diagnosed in the model output. $HNO_3$
formation from $NO_3$ + HC and the hydrolysis of $XNO_3$ are small globally (5-6%), but the latter is dominant over the
remote oceans (Figure 3).





Figures 4 - 6 show modeled $\Delta^{17}O$(nitrate) for the "cloud chemistry" simulation (the "standard" simulation is shown in
Figures S2 – S4). Figure 4 shows modeled annual-mean $\Delta^{17}O$(nitrate) below 1 km altitude. The model predicts an
annual-mean range of $\Delta^{17}O$(nitrate) = 4 – 33‰ near the surface. The lowest values are over Amazonia due to the
dominance of $RONO_2$ hydrolysis and the highest values are over the mid-latitude oceans due to the dominance of
$XNO_3$ hydrolysis (Figures 3 and 4).
Figure 5 compares the model with a global compilation of $\Delta^{17}O$(nitrate) observations from around the world.
Observations included in Figure 5 include locations where there is enough data to calculate monthly means at each
location (McCabe et al., 2006;Kunasek et al., 2008;Hastings et al., 2003;Kaiser et al., 2007;Michalski et al.,
2003;Guha et al., 2017;Savarino et al., 2013;Ishino et al., 2017b;Savarino et al., 2007;Alexander et al., 2009;He et al.,
2018b). Figure 6 compares the seasonality in modeled $\Delta^{17}O$(nitrate) to the observations where samples were collected
over the course of approximately one year (McCabe et al., 2006;Kunasek et al., 2008;Kaiser et al., 2007;Michalski et
al., 2003;Guha et al., 2017;Savarino et al., 2013;Ishino et al., 2017b;Savarino et al., 2007;Alexander et al., 2009). In
contrast to Alexander et al. (2009), the model does not significantly underestimate the $\Delta^{17}O$(nitrate) observations when
assuming $\Delta^{17}O(O_3)$ on the order of 25‰ (see Figure 2d in Alexander et al. (2009)). The increase in modeled
$\Delta^{17}O$(nitrate) is due to increased importance of $O_3$ in $NO_x$ cycling (85% below 1 km) compared to Alexander et al.
(2009) (80% below 1 km altitude), and an increase in the number and fractional importance of nitrate formation
pathways that yield relatively high values of $\Delta^{17}O$(nitrate) (red pathways in Fig. 1). Although XO species themselves
are only a minor NO oxidation pathway (1%), the addition of reactive halogen chemistry in the model has altered the
relative abundance of $O_3$ and $HO_x$ (Sherwen et al., 2016) in such a way as to increase the modeled $\Delta^{17}O(NO_x)$. The
Alexander et al. (2009) study used GEOS-Chem v8-01-01, which included tropospheric nitrate formation from the
NO + OH, $N_2O_5$ + $H_2O$, and $NO_3$ + HC pathways only. An increased importance of $N_2O_5$ hydrolysis (R4) and
additional nitrate formation pathways that yield relatively high values of $\Delta^{17}O$(nitrate) (R5, R6, R8, and R10) in the
present study also explain the increase in modeled $\Delta^{17}O$(nitrate) relative to Alexander et al. (2009). Assuming a value
of 35‰ for $\Delta^{17}O(O_3)$ in the model that did not include reactive halogen chemistry or heterogeneous reactions in cloud
water produced good agreement between modeled and observed $\Delta^{17}O$(nitrate) in Alexander et al. (2009); however, in
the current version of the model this isotopic assumption leads to a model overestimate at nearly all locations (Figure





S5). The "cloud chemistry" model shows somewhat better agreement with the observations ($R^2 = 0.51$ in Figure 5)
compared to the "standard" model ($R^2 = 0.48$ in Figure S3). Improved agreement with the observations occurs in the
mid- to high-latitudes (Figures 6 and S4) is due to addition of $N_2O_5$ hydrolysis in clouds (Figures 3 and S6).
The mean $\Delta^{17}O$(nitrate) value of the observations ($27.7 \pm 5.0‰$) shown in Figure 5 is not significantly different from
the modeled values at the location of the observations ($28.6 \pm 4.5‰$); however, the range of $\Delta^{17}O$(nitrate) values of
the observations ($10.9 - 40.6‰$) is larger than in the model ($19.6 - 37.6‰$). As previously noted in Savarino et al.
(2007), the maximum observed $\Delta^{17}O$(nitrate) value ($40.6‰$) is not possible given our isotope assumption for the
terminal oxygen atom of ozone ($\Delta^{17}O(O_3^*) = 39‰$). Observed $\Delta^{17}O$(nitrate) $> 39‰$ (in Antarctica) has been suggested
to be due to transport of nitrate from the stratosphere (Savarino et al., 2007), as stratospheric $O_3$ is expected to have a
higher $\Delta^{17}O(O_3)$ value than ozone produced in the troposphere (Krankowsky et al., 2000;Mauersberger et al.,
2001;Lyons, 2001). Indeed, the model underestimates the observations at Dumont d'Urville (DDU) and the South
Pole (both in Antarctica) during winter and spring (Figure 6), when and where the stratospheric contribution is
expected to be most important (Savarino et al., 2007). The model underestimate in Antarctica may also be due to
model underestimates of BrO column (Chen et al., 2017) and ozone abundance (Sherwen et al., 2016) in the southern
high latitudes. The largest model overestimates occur at Mt. Lulin, Taiwan (Figures 5 and 6). Based on nitrogen
isotope observations ($\delta^{15}N$), nitrate at Mt. Lulin is thought to be influenced by anthropogenic nitrate emitted in polluted
areas of mainland China and transported to Mt. Lulin, rather than local nitrate production (Guha et al., 2017). The
model compares well to the mid-latitude locations close to pollution sources (La Jolla and Princeton), although the
model underestimates winter time $\Delta^{17}O$(nitrate) in La Jolla, CA, USA.
**4. Model uncertainties**
The uncertainty in the two most important nitrate formation pathways, $NO_2 + OH$ and $N_2O_5$ hydrolysis, and their
impacts on $NO_x$ and oxidant budgets, have been examined and discussed elsewhere (Macintyre and Evans,
2010;Newsome and Evans, 2017;Holmes et al., 2019). The impacts of the formation and hydrolysis of halogen nitrates
on global $NO_x$ and oxidant budgets have also been previously examined (Sherwen et al., 2016). Here we focus on
three additional processes using a set of model sensitivity studies. First, we examine the importance of the third most



important nitrate production pathway on the global scale as predicted by the "standard" model, $NO_2$ aerosol uptake
and hydrolysis, and its implications for the global $NO_x$, nitrate, and oxidant budgets.  Second, we examine the role of
changing anthropogenic $NO_x$ emissions over a 15-year period (2000 to 2015) on the relative importance of the
formation of inorganic nitrate from the hydrolysis of organic nitrates.  Finally, we examine the role of aerosol nitrate
photolysis on the relative importance of different nitrate formation pathways.  The impact of aerosol nitrate photolysis
on $NO_x$ and oxidant budgets has been examined in detail elsewhere (Kasibhatla et al., 2018).
**4.1  Heterogeneous uptake and hydrolysis of $NO_2$**
Heterogeneous uptake of $NO_2$ to form $HNO_3$ and HONO is the third most important nitrate formation pathway in the
"standard" model on the global scale (Figure 1).  The reaction probability ($\gamma_{NO2}$) measured in laboratory studies ranges
between $10^{-8}$ to $10^{-4}$ depending on aerosol chemical composition (Lee and Tang, 1988;Crowley et al., 2010;Gutzwiller
et al., 2002;Yabushita et al., 2009;Abbatt and Waschewsky, 1998;Burkhart et al., 2015;Broske et al., 2003;Li et al.,
2018a;Xu et al., 2018). A value of $\gamma_{NO2} = 10^{-4}$ is used in the "standard" model, which is at the high end of the reported
range.  A molar yield of 0.5 for both $HNO_3$ and HONO formation is assumed in the model based on laboratory studies
and hypothesized reaction mechanisms (Finlayson-Pitts et al., 2003;Jenkin et al., 1988;Ramazan et al., 2004;Yabushita
et al., 2009).  However, both the reaction rate and mechanism of this reaction and its dependence on chemical
composition and pH is still not well understood (Spataro and Ianniello, 2014).
The "cloud chemistry" simulation uses a reaction probability formulation for aerosol uptake of $NO_2$ ($\gamma_{NO2}$) that
depends on aerosol chemical composition, ranging from $\gamma_{NO2} = 10^{-8}$ for dust to $\gamma_{NO2} = 10^{-4}$ for black carbon based on
recent laboratory studies (Holmes et al., 2019).  The updated $NO_2$ reaction probability results in a negligible (<1%)
importance of this reaction for nitrate formation, compared to 12% contribution in the "standard" model. The "cloud
chemistry" simulation significantly increases the fractional importance of $N_2O_5$ hydrolysis (from 28 to 41%, globally
below 1 km altitude) compared to the "standard" simulation, in part due to decreased competition from $NO_2$ hydrolysis
and in part due to increased $N_2O_5$ hydrolysis in clouds.  To evaluate the relative importance of competition from $NO_2$
hydrolysis and the addition of $N_2O_5$ hydrolysis in clouds, we perform a model sensitivity study that is the same as the
"standard" simulation but decreases the reaction probability of $NO_2$ hydrolysis on aerosol ($\gamma_{NO2} = 10^{-7}$), without adding
$N_2O_5$ hydrolysis in clouds.  Similar to the "cloud chemistry" simulation, using $\gamma_{NO2} = 10^{-7}$ renders $NO_2$ hydrolysis a





negligible nitrate formation pathway, and increases the relative importance of $N_2O_5$ hydrolysis from 28% to 37%.
This suggests that reduced competition from $NO_2$ hydrolysis is the main reason for the increased importance of $N_2O_5$
hydrolysis in the "cloud chemistry" simulation, though the addition of heterogeneous reactions on clouds also plays a
role.
$NO_2$ hydrolysis represents a significant source of HONO in the "standard" model simulation; the reduced $NO_2$ reaction
probability from $\gamma_{NO2} = 10^{-4}$ to $\gamma_{NO2} = 10^{-7}$ results in a reduction of HONO below 1 km altitude by up to 100% over
the continents, with relatively small (up to 1 ppb) changes in nitrate concentrations (Figure 7). The reduction in the
rate of heterogeneous $NO_2$ uptake leads to reductions in OH where this reaction was most important in the model
(over China and Europe) due to reductions in HONO, but leads to increases in OH elsewhere due to increases in ozone
(by up to a few ppb) resulting from small increases in the $NO_x$ lifetime due to a reduction in the $NO_x$ sink (Figure 8).
Similar changes in HONO are seen when comparing the "standard" and "cloud chemistry" simulation (not shown).
Increased importance of $N_2O_5$ hydrolysis in both the "cloud chemistry" simulation and the simulation without cloud
chemistry but with a reduced reaction probability for $NO_2$ hydrolysis increases modeled annual-mean $\Delta^{17}O$(nitrate)
by up to 3‰ in China where this reaction is most important. This improves model agreement with monthly-mean
observations of $\Delta^{17}O$(nitrate) in Beijing (He et al., 2018a) (Figures 5 and S3).
The product yields of $NO_2$ hydrolysis are also uncertain. Jenkin et al. (1988) proposed the formation of a water
complex, $NO_2 \cdot H_2O$, leading to the production of HONO and $HNO_3$. Finlayson-Pitts et al. (2003) and Ramazan et al.
(2004) proposed the formation of the dimer $N_2O_4$ on the surface, followed by isomerization to form $NO^+NO_3^-$.
Reaction of $NO^+NO_3^-$ with $H_2O$ results in the formation of HONO and $HNO_3$. Laboratory experiments by Yabushita
et al. (2009) suggested that dissolved anions catalyzed the dissolution of $NO_2$ to form a radical intermediate $X-NO_2^-$
(where X = Cl, Br, or I) at the surface followed by reaction with $NO_2(g)$ to form HONO and $NO_3^-$. These experiments
described above were performed at $NO_2$ concentrations much higher than exist in the atmosphere (10 – 100 ppm)
(Yabushita et al., 2009;Finlayson-Pitts et al., 2003;Ramazan et al., 2004). A laboratory study utilizing isotopically
labeled water to investigate the reaction mechanism suggested that the formation of HONO resulted from the reaction
between adsorbed $NO_2$ and $H^+$, while the formation of $HNO_3$ resulted from the reaction between adsorbed $NO_2$ and
$OH^-$, and did not involve the $N_2O_4$ intermediate (Gustafsson et al., 2009). Results from Gustafsson et al. (2009)



suggest an acidity-dependent yield of HONO and HNO₃, favoring HONO at low pH values. A recent study in the
northeast U.S. during winter found that modeled nitrate abundance was overestimated using a molar yield of 0.5 for
HONO and HNO₃, and the model better matched the observations of NO₂ and nitrate when assuming a molar yield of
1.0 for HONO (Jaeglé et al., 2018). Particles were acidic (pH < 2) during this measurement campaign (Guo et al.,
2017;Shah et al., 2018), which may favor HONO production over HNO₃.
We examine the potential importance of this acidity-dependent yield by implementing a pH-dependent product yield
in two separate sensitivity simulations, first using an NO₂ aerosol uptake reaction probability of $\gamma = 10^{-4}$ as in the
"standard" simulation and second with $\gamma_{NO2} = 10^{-7}$. The acidity-dependent yield for HONO and HNO₃ formation is
based on the laboratory study by Gustafsson et al. (2009). We use aerosol pH calculated from ISORROPIA II
(Fountoukis and Nenes, 2007) to calculate the concentration of [H⁺] and [OH⁻] in aerosol water. The yield of HONO
($\Upsilon_{HONO}$) from heterogeneous uptake of NO₂ on aerosol surfaces is calculated according to E3:
$$\Upsilon_{HONO} = \frac{[H^+]}{[H^+]+[OH^-]}$$                                      (E3)
where [H⁺] and [OH⁻] are in units of M. The yield of HNO₃ from this reaction is equal to (1 - $\Upsilon_{HONO}$). E3 yields values
of $\Upsilon_{HONO}$ near unity for aerosol pH values less than 6, decreasing rapidly to zero between pH values between 6-8
(Figure 9). Calculated aerosol pH values are typically < 6 in the model except in remote regions far from NOₓ sources
(Figure S7), favoring the product HONO.
The acidity-dependent yield implemented in the "standard" simulation with $\gamma_{NO2} = 10^{-4}$ increases HONO
concentrations by up to 1 ppbv in China where this reaction is most important (Figure 10). Fractional increases in
HONO exceed 100% in remote locations (Figure 10). Increased HONO leads to increases in OH on the order of 10
– 20% in most locations below 1 km altitude, while ozone concentrations increase in most locations by up to several
ppbv (Figure 10). The exception is the southern high latitudes; likely due to decreased formation and thus transport
of nitrate to remote locations. The impact on NOₓ and nitrate budgets is relatively minor. The global, annual mean
NOₓ burden near the surface (below 1 km) increases slightly (+2%) as a result of the decreased rate of conversion of
NO₂ to nitrate; the change to the global tropospheric burden is negligible. Annual-mean surface nitrate concentrations
show small decreases up to 1 ppbv in China where this reaction is most important in the model; impacts on nitrate
concentrations over a shorter time period may be more significant (Jaeglé et al., 2018). The fraction of HNO₃ formed



from NO$_2$ + OH (49%) increases due to increases in OH from the HONO source. The fraction of HNO$_3$ formation

from the uptake and hydrolysis of N$_2$O$_5$ also increases (from 28% to 32%) due to reductions in the nighttime source

of nitrate from NO$_2$ hydrolysis. The calculated mean $\Delta^{17}$O(nitrate) at the location of the observations shown in Figure

5 (27.9 ± 5.0‰) is not significantly impacted due to compensating effects from changes in both high- and low-

producing $\Delta^{17}$O(nitrate) values. Modeled monthly mean $\Delta^{17}$O(nitrate) in China, where NO$_2$ hydrolysis is most

important increases by ~1‰, but is still biased low by 1-2‰.

Using a combination of both the low reaction probability (γ = 10$^{-7}$) and the acidity-dependent yield gives similar results

as using γ = 10$^{-7}$ and assuming a molar yield of 0.5 for HONO and HNO$_3$ (not shown). In other words, including a

pH-dependent product yield rather than a yield of 0.5 for HONO and nitrate results in negligible differences for

oxidants, NO$_x$ and nitrate abundances when the reaction probability (γ$_{NO2}$) is low.

**4.2 Hydrolysis of organic nitrates (RONO$_2$)**

Anthropogenic NO$_x$ emissions have been increasing in China and decreasing in the U.S. and Europe (Richter et al.,

2005;Hoesly et al., 2018b), with implications for the relative importance of inorganic and organic nitrate formation as

a sink for NO$_x$ (Zare et al., 2018). To examine the impacts of recent changes in anthropogenic NO$_x$ emissions for

nitrate formation pathways, we run the "standard" model using the year 2000 emissions and meteorology after a 1-

year model spin up, and compare the results to the "standard" model simulation run in the year 2015. This time-period

encompasses significant changes in anthropogenic NO$_x$ emissions in the U.S., Europe, and China, and encompasses

most of the time period of the observations shown in Figures 5 and 6. Total, global anthropogenic emissions of NO$_x$

are slightly lower in the 2000-year simulation (30 Tg N yr$^{-1}$) compared to the year 2015 simulation (31 Tg N yr$^{-1}$) due

to decreases in North America and Europe, counteracted by increases in Asia (Figure S7). This leads to increases of

less than 10% in the annual-mean, fractional importance of the source of nitrate from the hydrolysis of organic nitrates

in the U.S., and corresponding decreases of less than 10% over China (Figure 11). Relatively small changes (< 10%)

in nitrate formation pathways yield small changes (< 2‰) in modeled annual-mean $\Delta^{17}$O(nitrate) between the year

2000 and 2015, differences in $\Delta^{17}$O(nitrate) over shorter time periods may be larger. Changes in the formation of

nitrate from the hydrolysis of RONO$_2$ remains unchanged globally, as increases in the U.S. and Europe and decreases

in China counteract one another.



### 4.3 Photolysis of aerosol nitrate

Observations have demonstrated that aerosol nitrate can be photolyzed at rates much faster than $HNO_3(g)$ (Reed et al., 2017;Ye et al., 2016); however, the magnitude of the photolytic rate constant is uncertain. We examine the implications of this process for global nitrate formation pathways by implementing the photolysis of aerosol nitrate as described in Kasibhatla et al. (2018) into the "standard" model simulation, scaling the photolytic rate constant for both fine- and coarse-mode aerosol nitrate to a factor of 25 times higher than that for $HNO_3(g)$ (Kasibhatla et al., 2018;Romer et al., 2018), with a molar yield of 0.67 for HONO and 0.33 for $NO_x$ production. The global, annual mean $NO_x$ burden near the surface (below 1 km) increases slightly (+2%) as a result of the photolytic recycling of nitrate to $NO_x$, similar to Kasibhatla et al. (2018). Aerosol nitrate photolysis results in only small impacts on the relative importance of nitrate formation pathways (< 2%) likely due to simultaneous increases in $O_3$ and OH (Kasibhatla et al., 2018), which in turn yields small impacts on calculated $\Delta^{17}O$(nitrate) at the location of the observations shown in Figure 5 (27.9 ± 5.0‰). Nitrate photolysis itself has minimal impact on $\Delta^{17}O$(nitrate) because it is a mass-dependent process (McCabe et al., 2005).

## 5    Conclusions

Observations of $\Delta^{17}O$(nitrate) can be used to help quantify the relative importance of different nitrate formation pathways. Interpretation of $\Delta^{17}O$(nitrate) requires knowledge of $\Delta^{17}O(O_3)$, which until recently was highly uncertain. Previous modeling studies showed good agreement between observed and modeled $\Delta^{17}O$(nitrate) when assuming $\Delta^{17}O(O_3) = 35‰$. However, recent observations of $\Delta^{17}O(O_3)$ from around the world have shown $\Delta^{17}O(O_3) = 26 ± 1‰$, suggesting that models are underestimating the role of ozone relative to $HO_x$ in $NO_x$ chemistry. We utilize a global compilation of observations of $\Delta^{17}O$(nitrate) to assess the representation of nitrate formation in a global chemical transport model (GEOS-Chem). The modeled $\Delta^{17}O$(nitrate) is roughly consistent with observations, with a mean modeled and observed $\Delta^{17}O$(nitrate) of (28.6 ± 4.5‰) and (27.6 ± 5.0‰), respectively, at the locations of the observations. Improved agreement between modeled and observed $\Delta^{17}O$(nitrate) is due to increased importance of ozone versus $HO_2$ and $RO_2$ in $NO_x$ cycling and an increase in the number and importance of nitrate production pathways that yield high $\Delta^{17}O$(nitrate) values. The former may be due to implementation of tropospheric reactive



halogen chemistry in the model, which impacts ozone and $HO_x$ abundances. The latter is due mainly to increases in the relative importance of $N_2O_5$ hydrolysis, with the hydrolysis of halogen nitrates also playing an important role in remote regions.

The main nitrate formation pathways in the model below 1 km altitude are from $NO_2 + OH$ and $N_2O_5$ hydrolysis (both 41%). The relative importance of global nitrate formation from the hydrolysis of halogen nitrates and hydrogen-abstraction reactions involving the nitrate radical ($NO_3$) are of similar magnitude (~5%). The formation of nitrate from the hydrolysis of organic nitrate has increased slightly in the U.S. and decreased in China (changes <10%) due to changing $NO_x$ emissions from the year 2000 to 2015, although the global mean fractional importance (6%) remains unchanged as the regional changes counteract one another. Nitrate formation via heterogeneous $NO_2$ and $NO_3$ uptake and $NO_2 + HO_2$ are negligible (<2%). Although aerosol nitrate photolysis has important implications for $O_3$ and OH, the impacts on nitrate formation pathways are small.

The model parameterization for heterogeneous uptake of $NO_2$ has significant impacts on HONO and oxidants (OH and ozone) in the model. HONO production from this reaction has been suggested to be an important source of OH in Chinese haze due to high $NO_x$ and aerosol abundances (Hendrick et al., 2014;Tong et al., 2016;Wang et al., 2017), with implications for the gas-phase formation of sulfate aerosol from the oxidation of sulfur dioxide by OH (Shao et al., 2018;Li et al., 2018b). More recent laboratory studies suggest that the reaction probability of $NO_2$ on aerosols is lower than that previously used in the model. Using an $NO_2$ reaction probability formulation that depends on the chemical composition of aerosols as described in Holmes et al. (2019) renders this reaction negligible for nitrate formation, and has significant implications for modeled HONO, ozone, and OH. Although uncertainty also exists in the relative yield of nitrate and HONO from this reaction, the impacts of this assumption are negligible when we use these updated $NO_2$ reaction probabilities. Observations of $\Delta^{17}O(nitrate)$ in Chinese haze events during winter (He et al., 2018b) may help to quantify the importance of this nitrate production pathway in a region where the model predicts it is significant.

Author contributions: B.A. designed the study and performed the model simulations and calculations. All other authors provided model code and contributed to writing and analysis.





**Acknowledgements**:
B.A. acknowledges NSF AGS 1644998 and 1702266 and helpful discussions with Joël Savarino and Ron Cohen.
C.D.H. acknowledges the NASA New Investigator Program grant NNX16AI57G.  J.A.F. acknowledges Australian
Research Council funding DP160101598.

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

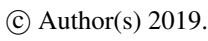



**Table 1.** Calculated $\Delta^{17}O$(nitrate) in the model for each nitrate production pathway (X = Br, Cl,
and I; HC = hydrocarbon; MTN = monoterpene; ISOP = isoprene; $\Delta^{17}O(O_3^*) = 39‰$).

|     | **Nitrate formation pathway** | **$\Delta^{17}$O(nitrate)** |
|-----|-------------------------------|------------------------------|
| \multicolumn{2}{c}{Gas-phase reactions} | |
| R1 | $NO_2$ + OH | $\frac{2}{3}A\Delta^{17}O(O_3^*)$ |
| R2 | $NO_3$ + HC | $\left(\frac{2}{3}A + \frac{1}{3}\right)\Delta^{17}O(O_3^*)$ |
| R3 | NO + $HO_2$ | $\frac{1}{3}A\Delta^{17}O(O_3^*)$ |
| \multicolumn{2}{c}{Aerosol uptake from the gas-phase followed by hydrolysis} | |
| R4 | $N_2O_5$ + $H_2O_{(aq)}$ | $\left(\frac{2}{3}A + \frac{1}{6}\right)\Delta^{17}O(O_3^*)$ |
| R5 | $N_2O_5$ + $Cl^-$(aq) | $\left(\frac{2}{3}A + \frac{1}{3}\right)\Delta^{17}O(O_3^*)$ |
| R6 | $XNO_3$ + $H_2O_{(aq)}$ | $\left(\frac{2}{3}A + \frac{1}{3}\right)\Delta^{17}O(O_3^*)$ |
| R7 | $NO_2$ + $H_2O_{(aq)}$ | $\frac{2}{3}A\Delta^{17}O(O_3^*)$ |
| R8 | $NO_3$ + $H_2O_{(aq)}$ | $\left(\frac{2}{3}A + \frac{1}{3}\right)\Delta^{17}O(O_3^*)$ |
| R9 | $RONO_2$ + $H_2O_{(aq)}$ (where $RONO_2$ is from NO + $RO_2$) | $\frac{1}{3}A\Delta^{17}O(O_3^*)$ |
| R10 | $RONO_2$ + $H_2O_{(aq)}$ (where $RONO_2$ is from $NO_3$ + MTN/ISOP) | $\left(\frac{2}{3}A + \frac{1}{3}\right)\Delta^{17}O(O_3^*)$ |





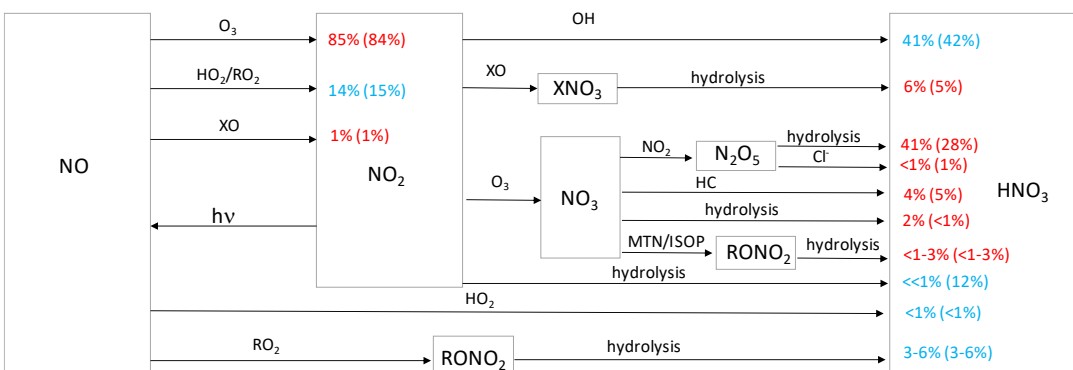

**Figure 1.** Simplified $HNO_3$ formation in the model.  Numbers show the global, annual mean percent
contribution to $NO_2$ and $HNO_3$ formation in the troposphere below 1 km for the "cloud chem"
("standard") simulation. Red indicates reactions leading to high $D^{17}O$ values, blue indicates reactions
leading to low $D^{17}O$ values.  $HO_2 = HO_2 + RO_2$; X = Br+Cl+I;  HC = hydrocarbons; MTN = monoterpenes;
ISOP = isoprene.

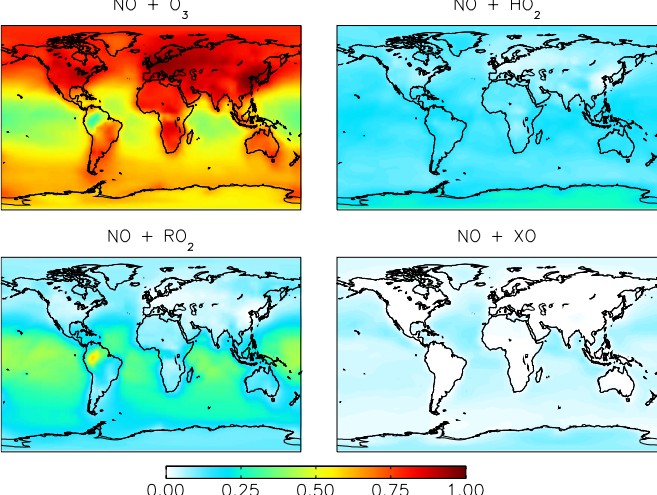

**Figure 2.** Annual-mean fraction of $NO_2$ formation from the oxidation of NO in the troposphere below 1
km altitude in the "cloud chemistry" model.




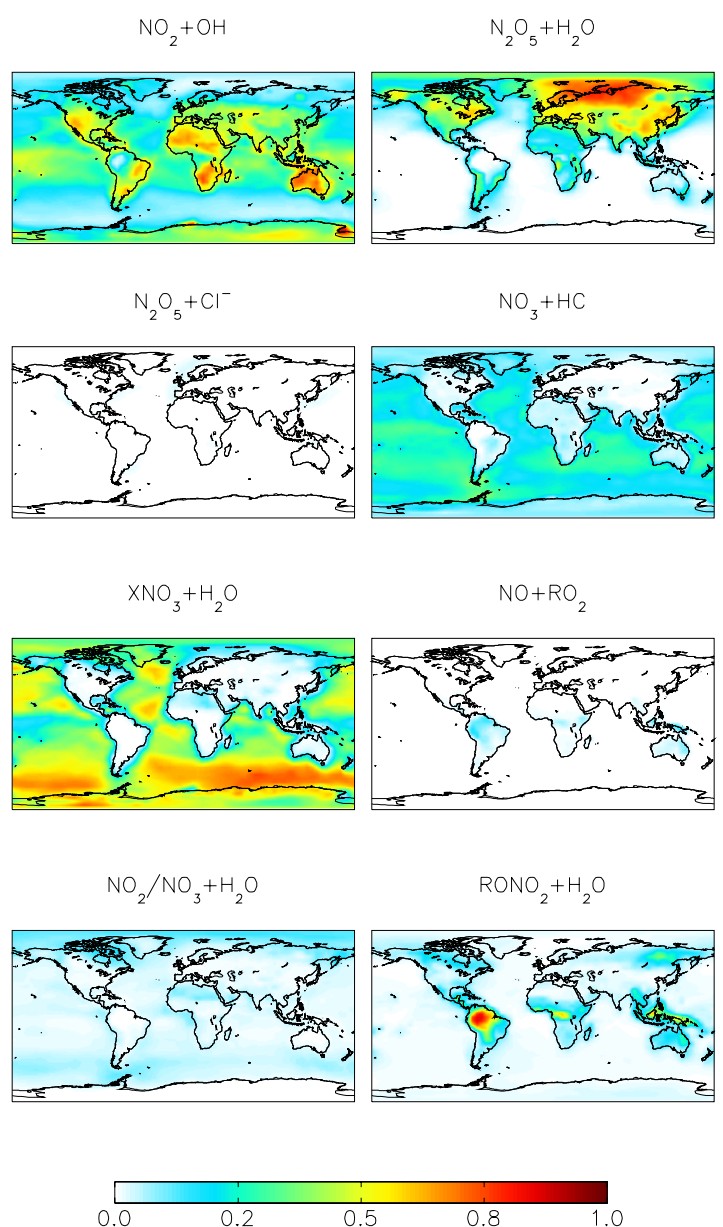

**Figure 3.** Annual-mean fraction of $HNO_3$ formation from the oxidation of $NO_x$ in the troposphere below 1
km altitude in the "cloud chemistry" model.



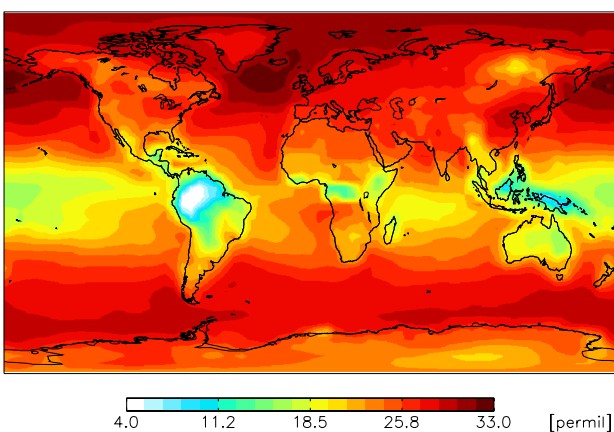

**Figure 4.** Modeled, annual-mean $\Delta^{17}O$(nitrate) below 1 km altitude for the "cloud chemistry" model.

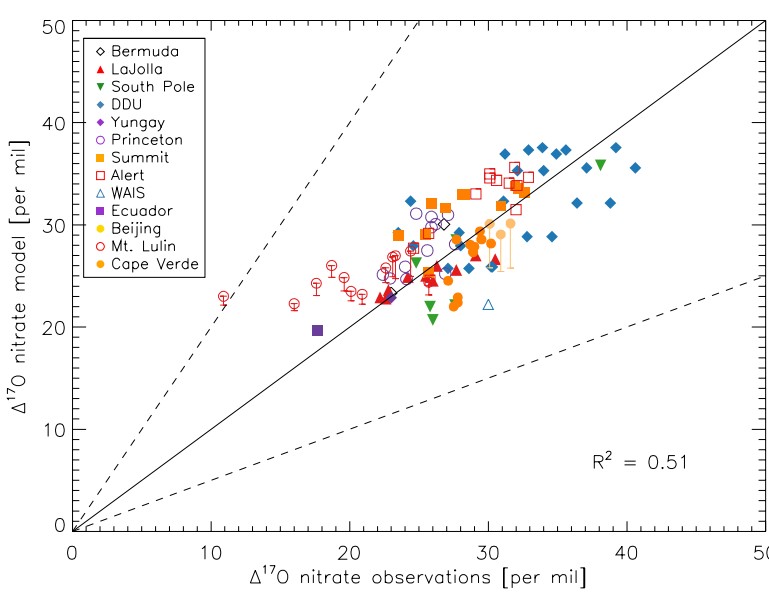

**Figure 5.** Comparison of monthly-mean modeled ("cloud chemistry") and observed $D^{17}O$(nitrate) at

locations where there are enough observations to calculate a monthly mean. References for the

observations are in the text. The error bars represent different assumptions for calculated modeled *A*

values for nighttime reactions as described in the text. Error bars for Beijing and Mt. Lulin reflect the

range of possible modeled *A* values for nighttime reactions as described in the text.





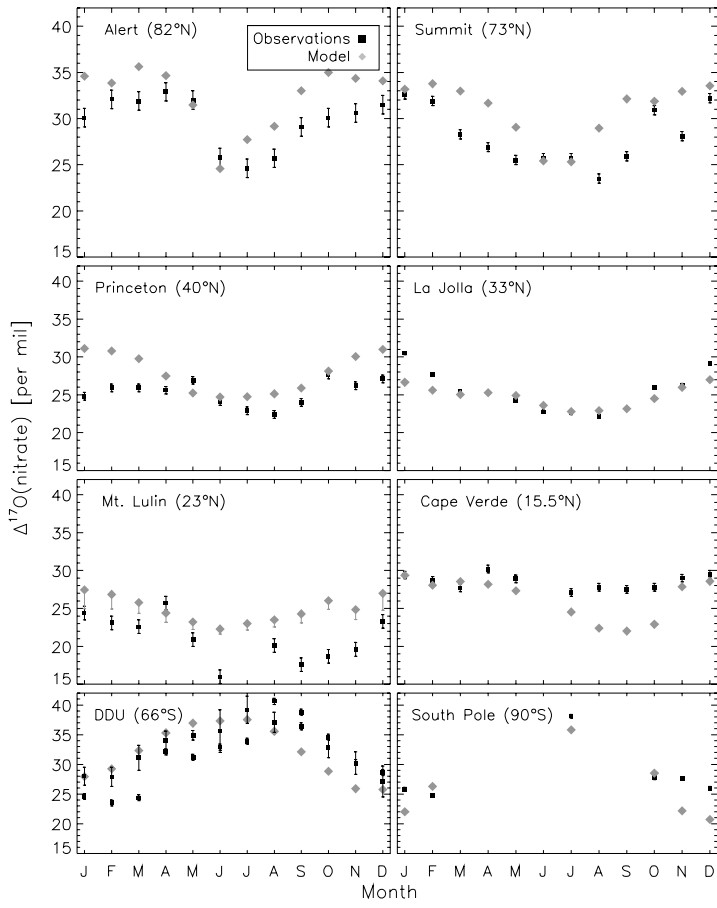

2 **Figure 6.** Comparison of monthly-mean modeled ("cloud chemistry") and observed $D^{17}O$(nitrate). Error

3 bars for Mt. Lulin reflect the range of possible modeled $A$ values for nighttime reactions as described in

4 the text.





**Figure 7.** Modeled annual-mean HONO (left) and fine-mode nitrate (right) concentrations below 1 km altitude in the "standard" simulation (top) with $g_{NO2} = 10^{-4}$ for $NO_2$ hydrolysis. Absolute (middle) and relative (bottom) change in concentrations below 1 km altitude between the "standard" model and the model simulation with $g_{NO2} = 10^{-7}$. Negative numbers represent a decrease relative to the standard simulation.





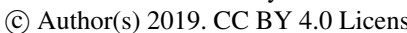

**Figure 8.** Same as Figure 7 except for OH (left) and ozone (right).



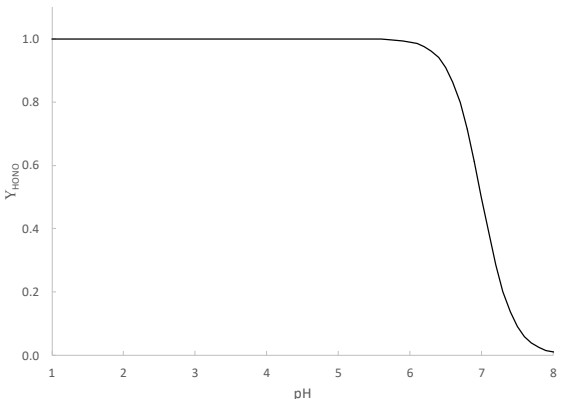

**Figure 9.** Calculated yield of HONO from the heterogeneous reaction of $NO_2$ on aerosol surfaces as a
function of pH.

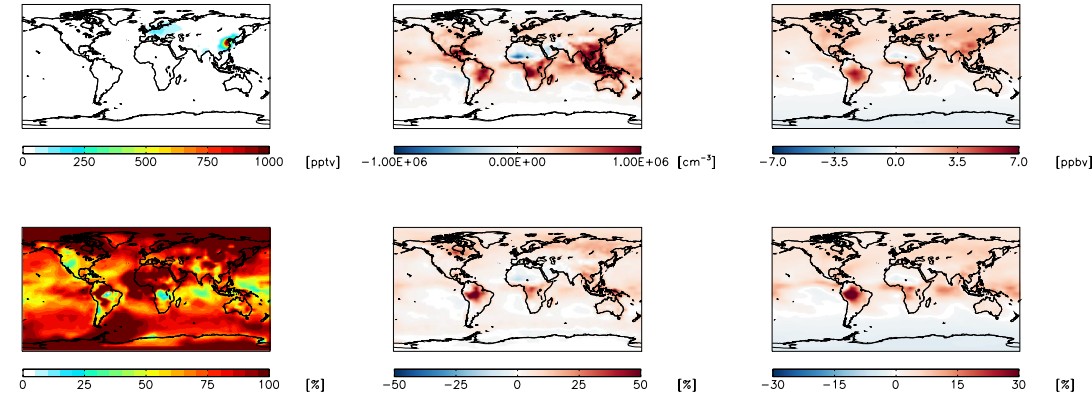

**Figure 10.** Absolute (top) and relative (bottom) change in HONO (left), OH (middle), and ozone (right)
concentrations below 1 km altitude between the "standard" model and the model simulation with an
acidity-dependent yield from $NO_2$ hydrolysis. Positive numbers represent an increase relative to the
"standard" simulation.



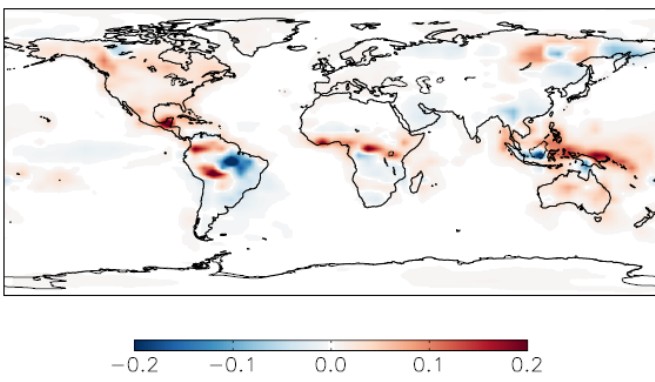

**Figure 11.** Modeled annual-mean difference in the fractional production rate of HNO₃ from the
hydrolysis of organic nitrate below 1 km attitude in the year 2015 relative to 2000 (2015 − 2000).

