# Peer review of "Global inorganic nitrate production mechanisms"

_Atmospheric Chemistry and Physics, 2019_

## Short Comment (SC1) · 20 May 2019

**A specific comment to Alexander et al.'s manuscript 'Global inorganic nitrate production mechanisms: Comparison of a global model with nitrate isotope observations'**

Certainly this is a comprehensive modelling study on global inorganic nitrate production mechanisms with a major aim of investigating how they affect global oxygen isotopic composition of nitrate. A state-of-the-art model (GEOS-Chem) is used in this study. Here what I want to address is the role of reactive halogens (BrO, ClO or IO) on the formation of nitrate. As mentioned in the manuscript, the hydrolysis of halogen nitrates ($XNO_3$, where X=Br, Cl, or I) is an important pathway for the inorganic nitrate formation, particularly in marine boundary layer, where open ocean sea spray serves as a large source of inorganic halogens. Basing on their modelling integrations, the authors conclude that halogens is not important and only accounts for ~6% of the global (<1 km) nitrate production. The dominant processes are reaction of $NO_2+OH$ and hydrolysis of dinitrogen pentoxide ($N_2O_5$), each accounting for 41% of the production respectively.

However, in an early tropospheric global model (p-TOMCAT) bromine study (Yang et al., 2005, Figure 12c), what they derived is just opposite: A month-long integration (March) shows that $BrNO_3$ hydrolysis reaction can cause a net reduction of lower tropospheric NOx (=NO+$NO_2$) by 40-80% at latitudes >50S in the Southern Hemisphere and by 20-60% at latitudes >70N in the Northern Hemisphere, though the reduction in the tropical regions is very small (<1%) (due to very lower BrO simulated and relatively higher OH concentrations). Note that this modelling work did include any sea ice sourced bromine source (an additional bromine source to the polar troposphere apart from sea spray and short-lived halocarbons). Thus the actual contribution from halogens could be even higher than the result shown in the paper. Then my question is why these two global models give such a big different result of the bromine-related NOx reduction (or $HNO_3$ production)? Please discus about it and supply more information such as surface layer BrO from the GEOS-Chem model form comparison. To help diagnose which halogen dominate, please supply each individual contribution (from Br, Cl and I) or tell clearly which halogen dominates the nitrate formation.

In addition, the values shown in Figure 3 of the manuscript really puzzle me. The annual fraction of $HNO_3$ formation from the oxidation of NOx in the troposphere below 1km altitude from the '$XNO_3+H_2O$' is almost at same level as the '$NO_2+OH$' and '$N2O_5+H_2O$' reactions. Why their global integration numbers are so different, e.g. by almost an order of magnitude, 6% vs 41%? Please explain it.

Reference:

Yang, X., R. A. Cox, N. J. Warwick, J. A. Pyle, G. D. Carver, F. M. O'Connor and N. H. Savage, Tropospheric bromine chemistry and its impacts on ozone: A model study. *J. Geophys. Res.*, 110, D23311, doi:10.1029/2005JD006244, 2005

---

## Referee Comment (RC1) · Anonymous Referee #1 · 2 Jun 2019

The authors present an interesting analysis of oxygen isotopes in nitrate as a constraint on global chemistry. They consider the main processes, the role of a variety of interesting more minor ones and conclude that N2O5 and OH driven processes are about equal contributors to the global average nitrate source.

I recommend publication as is.

---

## Referee Comment (RC2) · Anonymous Referee #2 · 8 Jun 2019

This work is a modeling study focused on the formation of atmospheric nitrate. Based upon the oxygen isotopic composition of nitrate, specifically D17O, the authors utilize the model to quantify the different production pathways of atmospheric nitrate. D17O is a useful tracer for this purpose because it is sensitive to oxidation pathways mediated by ozone or oxygen atoms from ozone. Overall this study represents an update to an earlier 2009 study by the same lead author, because the global atmospheric chemistry model used (GEOS-Chem) has had considerable updates to its chemistry since the earlier work. Because this represents a follow on to an earlier study, in some ways, the

work is not as novel. Still, it makes an interesting contribution to field and is certainly a benchmark by which other isotopic modeling studies of its kind will be compared. There are a number of issues in the manuscript that should be addressed and several suggestions that the authors should take into consideration before publishing the work.

Abstract: revisit the abstract after incorporating comments from all reviewers. Additionally, clarify the percentages of the different pathways – one page 2 lines 2-4 it, at first read, appears as if you are only talking about 41% + 41% + 6%. It would be useful to keep in mind 1) that the isotopic composition from ozone does not appear as certain as presented, and 2) that the global compilation of observations is still heavily biased towards the northern mid-latitudes. In the sentence ending on line 11, I suggest adding "on a global scale." at the end of the sentence.

Page 3, line 2: double check the wavelength and provide a reference (e.g., JPL); if memory serves this should be <400 nm.

Page 3, lines 11-15: citations should be provided for each of these pathways, or at least something that sums this up.

Page 3, line 17: It does not make sense to cite Alexander et al., 2009 here. The global lifetime is not presented in that work, nor is it expressly calculated in this current manuscript, which it should be. Note below too that there are a variety of statements in the manuscript that are inconsistent with this broad statement here, which also may or may not represent the lifetime actually found in GEOS-Chem.

Page 3, line 21: add "For example" before "the photolysis of NO3- in snow grains..."

Page 5, lines 5-12. This is a major suggestion – please introduce here a clear distinction between the bulk ozone isotopic value versus the terminal isotopic value. This distinction was not made well in Alexander et al 2009 – was 35‰ (O3)bulk or (O3)trans? Similar for Michalski et al. This is a critical distinction that comes up later in the manuscript. Further discussion and review of the differences in these assumptions

amongst studies would be a useful addition to this manuscript. Otherwise, the reader is left feeling that there is a much wider gap in knowledge than suggested in the current study.

The language regarding new O3 observations "around the globe" needs to be expanded upon and clarified. Three studies, using the same technique and largely averaging over vast stretches of the globe do not equal "around the globe".

This is a minor point, but please do consider that, while the newer observations are certainly more consistent than previous work, a detailed look at the methodology in Vicars et al. (RCM, 26, 1219-1231) shows that VERY large corrections are necessary for this method to yield the appropriate D17O(O3) results. It would behoove the authors of the current manuscript to consider whether they want to hang their hat on the absolute certainty of this new technique before it is, at the very least, used by other groups in laboratory and field studies.

Page 6, line 13: In Figure 1, NO2 is not shown to react with HO2. This should be OH?

Page 7, lines 21-23. Transport of NO3- is not considered in the model, such that the results will reflect the "locally" produced NO3-. Here it is suggested that this will make" little difference in polluted regions where most nitrate is formed locally." Evidence for this – from the model and/or from the literature – need to be included here. At first glance, this is inconsistent with the statement in the introduction that the average lifetime is 3 days.

Furthermore, as highlighted later, the actual results from the model do not agree well enough with observations to assume that the lack of transport is not important. Can the authors further comment on the potential bias this might cause, particularly for regions where long-range organic nitrate transport would be important?

Page 9, line 7: please further explain this equation, I simply do not understand it. Why is 0.25 simply added?

[Figure]

Page 9, line 19: the measurement work for D17O(O3) does present error (i.e. analytical reproducibility and differences from the average when combining all measurements "across the globe"). This should be expressly stated here. Further, this uncertainty should be discussed in the results and discussion in terms of how sensitive the final products are to the fact that D17O(O3) can vary by a couple of per mil.

Page 10, lines 1-2: It probably should be noted that many of the observations compared with are precipitation nitrate, and therefore not only representative of the surface. Perhaps here is could be stated how many datasets compared with represent surface aerosol collection versus precipitation? How important is this in the disagreement between the observations and model?

Page 11, line 15 and line 25: Is the $\Delta17O(O3)$ on the order of 25‰ representative of the bulk or transferrable component of O3? Again, a careful discussion of bulk versus terminal is warranted in this manuscript and should be made clear throughout when referring to the isotopic composition that is actually transferred to nitrate ultimately.

Page 11, lines 15-18: How much does the D17O(nitrate) increase? Can you elaborate further on this point about the increase in modeled nitrate due to increased importance of O3 in NOx cycling (85%) compared to the earlier 80%? Does this increase play a larger role than the post-NO2 reactions?

Page 12, lines 5-14: This section should also include comment on why observations of D17O have found lower values than produced by the model.

Page 12, line 7-14: This discussion is strange. The Savarino et al, 2007 work comes before the updated and much more certain (according to this manuscript) observation of D17O(O3). The error in $\Delta17O(O3^*)$ has been reported to be 39+/-2 per mil, which seems to indicate that a tropospheric value as high as 41‰ could be possible. Further, Savarino's later work (Vicars and Savarino, 2014 cited in the manuscript! and comments of Savarino himself in presentations and in discussions on ACPD) in fact negates this conclusion suggesting that the photolysis and reformation of stratospheric

ozone that enters the troposphere should reset the tropospheric ozone to local values (see discussion of this in Fibigier et al., JGR, 2016). Values near 40 per mil have also been observed in Greenland – by a different research group using different techniques – so it seems highly improbable that the values near 40 per mil simply cannot exist. Please update and reframe this discussion based upon more recent work and consider also acknowledging the observations in Greenland such as (Fibiger et al, JGR-Atmos., 121, 5010 5030, 2016) and references therein.

Page 12 Lines 18-20: A more detailed comparison between modeled values and values observed in mid-latitudes should be made. The model results do not match well with results in Princeton despite the authors claims. In fact, the model matches better with La Jolla than with Princeton, so it is not clear why La Jolla is highlighted here. Point to figure comparisons more specifically here (ie Figure #'s).

The time series comparisons overall are not nearly as impressive as the global, bunched, comparison. What needs to be done in the community to get this more right? The time series speak to a lot of inconsistency in making local assumptions. For instance, there are important differences in model vs obs in the winter/spring of Princeton, Mt. Lulin, and La Jolla (and this likely speaks to the fact that local versus transported nitrate could be important); and then the fall values at Princeton, Mt. Lulin and Cape Verde are all not captured at all. More care should be taken and a more full discussion of the model/obs comparison should be done.

Page 12-13: It would be useful to summarize here what impact the model uncertainties reported in the other works has on D17O(NO3-). Nowhere in the current work is the model compared to NOx or nitrate observations – only the isotopic composition of nitrate. So at least framing what uncertainties are important for consideration and the type of impact they would have on D17O(NO3-) seem important here.

Page 13, lines 9-11 and line 27-28: I am not clear here why the uncertainty in the gamma(N2O5) is not considered here? E.g., the work by Bertram and Thornton (Atmos. Chem. Phys., 9, 8351–8363, 2009) and Tham et al. (Atmos. Chem. Phys., 18, 13155–13171, 2018) that suggest uptake coefficients range a lot from 0.001-0.035 and 0.005-0.039, respectively. How much uncertainty in D17O(NO3-) would changes in this parameter yield?

Page 16, Section 4.2: it should be made clear here why the "standard" simulation is used for this on not the cloud chemistry simulation, the latter is treated as if it is the state of the art through the rest of the manuscript.

This section (and the previous) is really interesting. The authors should consider adding in figures of change in D17O(NO3-) based on the sensitivity studies. The emphasis is placed on gas phase chemistry changes in the figures, which is interesting, but since the paper is really about D17O(NO3-) it seems a missed opportunity to show some change in D17O. This is especially important in that the comparison with the time series observations (Figure 6) is underserved in the manuscript and makes the model seem much more uncertain. Regional digestion might speak to why they are such inconsistencies in seasonality at several stations in the mid-lats OR it might speak to how much difference in D17O is not captured by not having transported nitrate. Furthermore, future observational studies that compare with this work will be 1) better served, 2) this work will be more cited, and 3) this will advance the community forward in terms of our understanding of atmospheric chemistry based upon D17O (i.e. where we understand it and where we don't!).

Page 16, lines 26-28: Globally, the D17O of nitrate remains relative unchanged from 2000 to 2015 emissions, but nitrate is not globally mixed. A more detailed regional analysis again would be really interesting here. For example, how does decreased NOx emissions impact the modeled D17O (and oxidation chemistry) and how does increased NOx emissions alter D17O in China? What are the implications for future observations?

Page 17, Conclusions: I again stress that the authors should make a full discussion of

bulk versus transferred isotopic anomaly and the implications of previous assumptions. While it is compelling that the global model agrees better with the updated bulk and transferred value (and note that the transferred values reported by Vicars have an uncertainty of 2 per mil!), the global model still only explains 51% of the variance and the time series plots by location show important and significant disagreements. In other words, it is not a convincing assumption that because the global agreement is better with 25 per mil as the bulk that the observations are correct.

As suggested above, having some discussion of regionality and figures of change in D17O(NO3) based upon the sensitivity studies would be useful, especially for observational work to compare with the model results and make progress on our understanding of key oxidation pathways.

This is minor, but it might be useful to acknowledge key assumptions in the work here and acknowledge what important steps forward are needed. Otherwise I imagine there will be a paper in another 10 years that will tell us "actually now we really know even better what we're doing" in atmospheric chemistry models. For instance, some key assumptions include: nitrate is locally produced; transported NO3- is not considered/treated nor is there any acknowledgement of how much of a difference this could make (see time series diffs!); NO emitted at night contains one-half its original O and one-half from local oxidant; the D17O(NO2) is calculated using 24-hour production rates [this is an improvement over earlier work but also means the production rates are out of sync with the nighttime versus daytime calculations]; model is compared with observations based on surface only.

Page 18, line 11: NO2+HO2 again? This not happening in the model correct?

Table 1: define A or refer directly to equation in text.

Figure 1: Consider that comparison (in the text) to the NO oxidation branching ratios of Alexander et al., (2009) would be interesting to allow for an understanding on how model updates have changed the modeled branching ratios with implications for D17O.

[Figure]

Please make D's symbols in the fig caption.

Figure 2: Adding an image of the D17O of NO2 would be interesting too.

Figure 5: I'm not sure +/- 50% is really appropriate for this figure (also these are not identified in the figure caption). If the model were more than +/-25% off we could consider it completely not in the right world! It might be interesting to add the best fit line from Alexander et al. (2009) to compare with present study. References for the observational studies should be explicit in the figure or make a table and refer to that table.

Figure 6: Again, it might be interesting to compare the time series with a what was predicted by the 2009 model. Reference for the observations need to be made. Delta should be a symbol. Why are there more than one observational point for DDU?

Figure 7: Gammas should be symbols.

Figure 9: this is not particularly useful – it is exactly described in the text, could be moved to supplement.

Figure 10: Not sure "acidity-dependent yield" is how it is referred to in the main text? It is difficult to see these figures in this format. For Figs 7-11 I found myself wanting to understand how much change in D17O there would be associated with different regions.

Supplement: Which simulation is used to produce Figure S1? Probably should make this explicit for all figures, or at least when it is NOT the cloud chemistry simulation. Is it possible to extend the color bar? It is difficult to digest since so much of the globe ends up close to 2 days.

Figure S2-S6: suggest including a more complete caption stating that this is . . . then same as. . .or for comparison with Figure...

Figure S5 caption is incorrect?

[Figure]

---

## Referee Comment (RC3) · Greg Michalski (Referee) · 17 Jul 2019

The discussion on A values on page 8 has a serious flaw, namely it ignores the rapid isotopic exchange between NO and NO2 (Sharma) and N2O5. This means that the $\Delta 17O = 0$ NO emitted at night does not have to be oxidized into NO2 to dilute NO2 $\Delta 17O$ value, but can simply exchange with existing NO2. Likewise, nighttime equilibrium NO3+NO2$<\longleftarrow\longrightarrow$ N2O5 would ultimately incorporate additional ozone into NO2. In other words there is a serious limitation to the counting oxidations and ignoring the exchanges during the nighttime. It probable that that at night isotope exchange equilibrium results in $\Delta 17O$ of NO = NO3 = NO2 This in turn would impact HONO $\Delta 17O$ and

NO2 "cloud chemistry" at night and HNO3 production early morning when O3 levels are low due to nighttime titration.

The other serious limitation is the treatment of the ozone $\Delta 17O$ value. It is well known that $\Delta 17O$ and $\delta 18O$ in ozone is a strong function of temperature and pressure. The choice of Vicars (Over cryogenic collection studies) because of the apparent constant $\Delta 17O$ values is because these were all surface measurements at effectively the same pressure and a narrow temperature range. It is unlikely O3 being recycled above the boundary layer will have a 26 per mil $\Delta 17O$. How much nitrate is formed in the mixed layer versus free troposphere? Also the authors have chosen to ignore our Atmos. Chem. Phys., 14, 4935–4953, 2014 paper where we showed the pressure and temperature dependence in NO2 $\Delta 17O$ values in equilibrium with O3 as a function of temperature and pressure that demonstrates this effect. A lot hinges on the validity of "Recently, much more extensive observations of ïAĎ17O(O3) using a new technique (Vicars et al., 2012) show ïAĎ17O(O3) = 26 $\pm$ 1‰ around the globe (Vicars et al., 2012;Ishino et al., 2017b;Vicars and Savarino, 2014), and suggest that previous modeling studies are biased low in ïAĎ17O(nitrate) (e.g., Alexander et al. (2009)), which would occur if the model underestimated the relative role of ozone in NOx chemistry."

These are nearly all clean marine boundary layer measurements and simply ignoring the Johnston and Krankowsky cryogenic collection is polluted urban environments seems to be cherry picking the data. Likewise our experimental NO2 $\Delta 17O$ values match well with that predicted by the T and P dependence of O3 formation experiments (i.e Thiemens, Mauersberger group). This is not the first paper to ignore these unpleasant contradictions. It seems no ones wants to acknowledge that something we do not understand is going on with either tropospheric O3 $\Delta 17O$ dynamics or their measurements.

Also they might include Wang et al. for some additional south American data https://www.sciencedirect.com/science/article/pii/S0016703714001811?via%3Dihub

---

## Author Comment (AC1) · 20 Sep 2019

**Reviewer comments in bold**, author responses in plain text.

**Certainly this is a comprehensive modelling study on global inorganic nitrate production mechanisms with a major aim of investigating how they affect global oxygen isotopic composition of nitrate. A state-of-the-art model (GEOS-Chem) is used in this study. Here what I want to address is the role of reactive halogens (BrO, ClO or IO) on the formation of nitrate. As mentioned in the manuscript, the hydrolysis of halogen nitrates (XNO3, where X=Br, Cl, or I) is an important pathway for the inorganic nitrate formation, particularly in marine boundary layer, where open ocean sea spray serves as a large source of inorganic halogens. Basing on their modelling integrations, the authors conclude that halogens is not important and only accounts for ~6% of the global (<1 km) nitrate production. The dominant processes are reaction of NO2+OH and hydrolysis of dinitrogen pentoxide (N2O5), each accounting for 41% of the production respectively.**

**However, in an early tropospheric global model (p-TOMCAT) bromine study (Yang et al., 2005, Figure 12c), what they derived is just opposite: A month-long integration (March) shows that BrNO3 hydrolysis reaction can cause a net reduction of lower tropospheric NOx (=NO+NO2) by 40-80% at latitudes >50S in the Southern Hemisphere and by 20-60% at latitudes >70N in the Northern Hemisphere, though the reduction in the tropical regions is very small (<1%) (due to very lower BrO simulated and relatively higher OH concentrations). Note that this modelling work did include any sea ice sourced bromine source (an additional bromine source to the polar troposphere apart from sea spray and short-lived halocarbons). Thus the actual contribution from halogens could be even higher than the result shown in the paper. Then my question is why these two global models give such a big different result of the bromine-related NOx reduction (or HNO3 production)? Please discus about it and supply more information such as surface layer BrO from the GEOS-Chem model form comparison. To help diagnose which halogen dominate, please supply each individual contribution (from Br, Cl and I) or tell clearly which halogen dominates the nitrate formation.**

A detailed description of the reactive bromine (and iodine) chemistry in the version of the model used in this study can be found in Sherwen et al., ACP, 2016.  We have not made any further modifications to the reactive halogen chemistry for this paper.  There is no conflict between this work and Yang et al. (2005).  Both Yang and Sherwen show that halogens have a large impact on $NO_x$ levels in the remote marine atmosphere. However, $NO_x$ levels and nitrate production in these regions are small regardless of the halogen chemistry, so halogens ($XNO_3$ hydrolysis) have a modest impact on global nitrate production, as shown here.

Sherwen et al. (2016) compared model results with and without reactive halogen chemistry. They found that the global, annual tropospheric $NO_x$ burden decreased by 3.1% due to $NO_x$ loss from the hydrolysis of $XNO_3$.  $ClNO_3$ and $BrNO_3$ hydrolysis were approximately equal contributors, while $INO_3$ was minor.  I cannot find a similar value for the impact of reactive halogens on global, annual tropospheric $NO_x$ in Yang et al. (2005) for direct comparison.  Yang

et al. (2005) state that $BrNO_3$ hydrolysis accounts for up to 60-80% of $NO_x$ loss at high latitudes, but it is much smaller (a few percent) at low latitudes. Figure 18 in Sherwen et al. (2018) suggests a similar spatial pattern, with $NO_x$ reductions up to ~80% in the high latitudes, and much smaller impacts in the low latitudes. Based on this, the results from these two models do not seem inconsistent. The Sherwen et al. (2016) results are also consistent with previous studies (Long et al., 2014; von Glasow et al., 2004; Parrella et al., 2012; Schmidt et al., 2016).

Sherwen et al. (2016) found that the model underestimates the tropospheric BrO column in high latitudes, especially in the southern hemisphere (see Figure 9 from Sherwen et al. (2016)). This is mentioned in the manuscript as a possible explanation for why the model underestimates $\Delta^{17}O$(nitrate) at high latitudes.

**In addition, the values shown in Figure 3 of the manuscript really puzzle me. The annual fraction of HNO3 formation from the oxidation of NOx in the troposphere below 1km altitude from the 'XNO3+H2O' is almost at same level as the 'NO2+OH' and 'N2O5+H2O' reactions. Why their global integration numbers are so different, e.g. by almost an order of magnitude, 6% vs 41%? Please explain it.**

Figure 3 shows the *fractional* importance of nitrate production pathways. $XNO_3$ hydrolysis is a dominant nitrate production pathway relative to the other pathways over the remote oceans in the mid to high latitudes. However, $NO_x$ emissions are pretty small in these regions, so that the contribution to total, global nitrate production in these remote oceanic regions is small. In the main (anthropogenic) $NO_x$ source regions, the $NO_2$ + OH and $N_2O_5$ hydrolysis pathways dominate local nitrate production, resulting in these reactions being dominant globally.

Although previous studies have not specifically reported the importance of $XNO_3$ hydrolysis for nitrate production (they focus instead on the importance for $NO_x$ loss), Sherwen et al. (2016) state that the rate of nitrate production from $XNO_3$ hydrolysis proceeds at a rate of 10% of $NO_x$ loss though the $NO_2$ + OH pathway. This seems consistent with results from the present study that $NO_2$ + OH is 41% of global nitrate production near the surface and $XNO_3$ hydrolysis is about 6%. We have added the following sentence to section 3:

"Although $XNO_3$ hydrolysis is the dominant nitrate formation pathway over the remote oceans (Figure 3), its contribution to total, global nitrate production is relatively small due to small local $NO_x$ sources in these regions."

---

## Author Comment (AC2) · 20 Sep 2019

Thank you for your positive comments.
* * *

---

## Author Comment (AC3) · 20 Sep 2019

**Reviewer comments in bold**, author responses in plain text.

**Abstract: revisit the abstract after incorporating comments from all reviewers.**

We have revised the abstract in response to reviewer #3 and to your comments below.

**Additionally, clarify the percentages of the different pathways – one page 2 lines 2-4 it, at first read, appears as if you are only talking about 41% + 41% + 6%.**

We added "individually" in this section of the abstract to clarify that 6% is not the sum of all the other pathways but represents the maximum contribution from each individual pathway. The sentence now reads:
"All other nitrate production mechanisms *individually* represent less than 6% of global nitrate production near the surface, but can be dominant locally."

**It would be useful to keep in mind 1) that the isotopic composition from ozone does not appear as certain as presented, and 2) that the global compilation of observations is still heavily biased towards the northern mid-latitudes. In the sentence ending on line 11, I suggest adding "on a global scale." at the end of the sentence.**

The phrase "on the global scale" has been added to the end of the abstract as suggested.

**Page 3, line 2: double check the wavelength and provide a reference (e.g., JPL); if memory serves this should be <400 nm.**

Correct, this has been changed to 398 nm based on the IUPAC recommendation.

**Page 3, lines 11-15: citations should be provided for each of these pathways, or at least something that sums this up.**

*Atkinson* [2000] sums this up nicely and has been cited.

**Page 3, line 17: It does not make sense to cite Alexander et al., 2009 here. The global lifetime is not presented in that work, nor is it expressly calculated in this current manuscript, which it should be. Note below too that there are a variety of statements in the manuscript that are inconsistent with this broad statement here, which also may or may not represent the lifetime actually found in GEOS-Chem.**

The *Park et al.* [2004] reference has been cited here instead. We have also added the following sentence to the end of the first paragraph of section 3:

"In the model, the global, annual mean lifetime of $NO_x$ in the troposphere against oxidation to nitrate is about 1 day; about 50% of this loss is from the reaction of $NO_2 + OH$. $NO_x$ loss from $N_2O_5$ becomes more important near the surface where aerosol surface area is relatively high. The global, annual mean lifetime of nitrate in the troposphere against wet and dry deposition to the surface is about 3 days."

**Page 3, line 21: add "For example" before "the photolysis of NO3- in snow grains. . ."**

Done. Thanks for this suggestion.

**Page 5, lines 5-12. This is a major suggestion – please introduce here a clear distinction between the bulk ozone isotopic value versus the terminal isotopic value. This distinction was not made well in Alexander et al 2009 – was 35‰ (O3)bulk or (O3)trans? Similar for Michalski et al. This is a critical distinction that comes up later in the manuscript. Further discussion and review of the differences in these assumptions amongst studies would be a useful addition to this manuscript. Otherwise, the reader is left feeling that there is a much wider gap in knowledge than suggested in the current study.**

We added here that we are referring to the bulk isotopic composition of ozone. The sentence now reads:
"Previous modeling studies showed good agreement with observations of $\Delta^{17}O$(nitrate) when assuming *that the bulk oxygen isotopic composition of ozone* ($\Delta^{17}O(O_3)$) is equal to 35‰."
Later in the manuscript (methods section) we present the distinction between bulk and terminal O-atom isotopic composition, where we define the terminal O-isotopic composition as $\Delta^{17}O(O_3*)$, as has been done in previous publications. Throughout the manuscript, we have changed the "*" symbol from a superscript font to regular font so it is easier for the reader to see. We also redefine the $\Delta^{17}O(O_3)$ symbol in the conclusions section.

**The language regarding new O3 observations "around the globe" needs to be expanded upon and clarified. Three studies, using the same technique and largely averaging over vast stretches of the globe do not equal "around the globe".**

Good point. The wording "around the globe" has been removed from the introduction. We double checked and this is the only time this term was used in the manuscript.

**This is a minor point, but please do consider that, while the newer observations are certainly more consistent than previous work, a detailed look at the methodology in Vicars et al. (RCM, 26, 1219-1231) shows that VERY large corrections are necessary for this method to yield the appropriate D17O(O3) results. It would behoove the authors of the current manuscript to consider whether they want to hang their hat on the absolute certainty of this new technique before it is, at the very least, used by other groups in laboratory and field studies.**

Thank you for this comment. Reviewer #3 also had this same concern. We have changed the following sentence:

"Reduction in uncertainty in the value of $\Delta^{17}O(O_3)$ enables improved interpretation of $\Delta^{17}O$(nitrate) as an observational constraint for the relative importance of nitrate formation pathways in the atmosphere."
to:
"These new observations of $\Delta^{17}O(O_3)$, combined with improved understanding and hence more comprehensive chemical representation of nitrate formation in models, motivates an updated comparison of observed and modeled $\Delta^{17}O$(nitrate) as an observational constraint for the relative importance of nitrate formation pathways in the atmosphere."

We have also changed wording on the value of $\Delta^{17}O(O_3)$ in the abstract, introduction, and conclusions so as not to imply that there is no remaining uncertainty in its value.

**Page 6, line 13: In Figure 1, NO2 is not shown to react with HO2. This should be OH?**

I assume you are referring to the reaction $NO + HO_2$ to form $HNO_3$? This is a termolecular reaction that is in competition with the bimolecular reaction $NO + HO_2 \rightarrow NO_2 + OH$. The branching between the termolecular and bimolecular reactions is such that less than 1% proceeds via the termolecular pathway. Hence, the termolecular reaction is often ignored. However, since it is included in the GEOS-Chem chemical mechanism, I show it in Figure 1. Figure 1 shows that this reaction is negligible. Kinetic data for these reactions can be found in IUPAC. We have added the following sentence to the methods section:

"The reaction of $NO + HO_2$ can also form $HNO_3$ directly, although the branching ratio for this pathway is < 1% (Butkovskaya et al., 2005)."

**Page 7, lines 21-23. Transport of NO3- is not considered in the model, such that the results will reflect the "locally" produced NO3-. Here it is suggested that this will make" little difference in polluted regions where most nitrate is formed locally." Evidence for this – from the model and/or from the literature – need to be included here. At first glance, this is inconsistent with the statement in the introduction that the average lifetime is 3 days.**

This is difficult to quantify without comparing model simulations with and without transporting the isotopic tracers. One would expect the highest bias in regions without a local source of $NO_x$, i.e., where all nitrate is formed elsewhere and transported to remote locations. Conversely, one would expect less bias in regions with strong local sources of $NO_x$ and hence nitrate production. However, since we do not quantify this bias, I changed "little" to "less". The sentence now reads:
"This should make *less* difference in polluted regions where most nitrate is formed locally."

**Furthermore, as highlighted later, the actual results from the model do not agree well enough with observations to assume that the lack of transport is not important. Can the authors further comment on the potential bias this might cause, particularly for regions where long-range organic nitrate transport would be important?**

Long-range transport of organic nitrates such as PAN would effectively represent a local source of $NO_x$ to remote regions upon decomposition to $NO_x$. Any other source of $NO_x$ that is effectively recycled, such as the photolysis of nitrate on snow grains, would also represent a local source of $NO_x$ and reduce the model bias resulting from lack of transport. To that end, we have further modified the previously mentioned sentence to the following to show that polluted areas are not the only regions where one might expect local source of $NO_x$ to dominate the source of nitrate:

"This should make less difference in polluted regions where most nitrate is formed locally, or for example in polar regions in summer when photochemical recycling of nitrate in the snowpack represents a significant local source of $NO_x$ at the surface."

We note that we don't state that transport is not important for simulating $\Delta^{17}O$(nitrate) at any particular location. Our approach without transporting the isotopic tracers will reflect the full range of calculated $\Delta^{17}O$(nitrate) on the global scale for any particular isotopic assumption. To be sure not to unintentionally imply that lack of transport is not a concern,wWe have modified this sentence to the following:

"Although lack of transport of the isotope tracers hinders direct comparison of the model with observations at any particular location, this approach will reflect the full range of possible modeled $\Delta^{17}O$(nitrate) values for the current chemical mechanism, which can then be compared to the range of observed $\Delta^{17}O$(nitrate) values around the globe."

In addition, we have elaborated on the potential role of lack of transport at particular locations in our extended discussion of comparison of the model with the locations shown in Figure 6.

**Page 9, line 7: please further explain this equation, I simply do not understand it. Why is 0.25 simply added?**

The value of 0.25 is a result of our low-end assumption of $A_{night}$ = 0.5, and that half of nitrate formed during the nighttime originates from NO emitted during that same night. The equation now reads:

"$A_{low} = 0.5A + 0.5A_{night}$, where $A_{night} = 0.5$"

**Page 9, line 19: the measurement work for D17O(O3) does present error (i.e. analytical reproducibility and differences from the average when combining all measurements "across the globe"). This should be expressly stated here. Further, this uncertainty should be discussed in the results and discussion in terms of how sensitive the final products are to the fact that D17O(O3) can vary by a couple of per mil.**

I think you are referring here to the standard deviation of the $\Delta^{17}O(O_3)$ observations, which is 1‰. This leads to an uncertainty of less than 1.5‰ in the calculated values of nitrate. We hesitate to add discussion of error bars based on this observed standard deviation (beyond stating it in the manuscript) because we don't want to suggest this represents a significant contribution to uncertainty in modeled values of $\Delta^{17}O$(nitrate). Indeed, some (see reviewer #3) suggest that these observations are biased low on the order of 5‰. To represent the full range

of likely possible $\Delta^{17}O(O_3)$ values, we also show a comparison of model results to observations when assuming a value of $\Delta^{17}O(O_3)$ that is at the high end (35‰) of the possible range based on observations, laboratory studies, and model simulations.  This figure is shown in the SI (Fig. S6).

**Page 10, lines 1-2: It probably should be noted that many of the observations compared with are precipitation nitrate, and therefore not only representative of the surface.**
**Perhaps here is could be stated how many datasets compared with represent surface aerosol collection versus precipitation? How important is this in the disagreement between the observations and model?**

We changed this sentence:
"We focus on model results near the surface because these can be compared to observations; currently only surface observations of $\Delta^{17}O(nitrate)$ are available."
to:
"We focus on model results near the surface (below 1 km) because these can be compared to observations; currently only surface observations of $\Delta^{17}O(nitrate)$ are available.  We note that two observation data sets (from Bermuda (Hastings et al., 2003) and Princeton, NJ (Kaiser et al., 2007)) are rainwater samples and thus may represent nitrate formed aloft.  However, since cloud water peaks on average near 1 km altitude in the MERRA2 meteorology used to drive GEOS-Chem, our model sampling strategy should capture the majority of the influence of clouds on nitrate formation."

**Page 11, line 15 and line 25: Is the _17O(O3) on the order of 25‰ representative of the bulk or transferrable component of O3? Again, a careful discussion of bulk versus terminal is warranted in this manuscript and should be made clear throughout when referring to the isotopic composition that is actually transferred to nitrate ultimately.**

We clarify in this sentence that we are referring to the bulk isotopic composition of ozone.

**Page 11, lines 15-18: How much does the D17O(nitrate) increase? Can you elaborate further on this point about the increase in modeled nitrate due to increased importance of O3 in NOx cycling (85%) compared to the earlier 80%? Does this increase play a larger role than the post-NO2 reactions?**

A back of the envelope calculation suggests that calculated $\Delta^{17}O(nitrate)$ would need to increase between 7 - 13.5‰ in order to explain why we got good agreement assuming a bulk ozone isotopic composition of 35‰ in the 2009 paper compared to only needing to assume 26‰ in the present paper.  The value of 13.5‰ is from the difference between 35‰ and 26‰ (9‰) times 1.5.  The upper limit (13.5‰) is assuming that all O-atoms come from ozone ($A = 1$ and all nitrate from R2).  The lower limit is assuming a lower end value of $A = 0.4$ (from Figure 2) and all nitrate forms from R1.  The actual difference is between these two end members, suggesting a difference on the order of 10‰.  On average, the increase in the value of $A$ from 0.80 to 0.85 would result in a difference of 0.05 * 39‰ = 2‰. This suggests, that on average,

the main difference is due to the increase in R2, R4, R5, and R6, although there is likely some temporal and spatial variability.  We have added the following sentence to address this:
"An increase in the average *A* value from 0.80 to 0.85 would tend to increase the calculated $\Delta^{17}O$(nitrate) on the order of 2‰ ($0.05 \times \Delta^{17}O(O_3{}^*)$), suggesting that the increase in the relative importance of the terminal reactions R4, R5, R6, R8, and R10 explains the majority of the difference between the results presented here compared to Alexander et al. (2009)."

**Page 12, lines 5-14: This section should also include comment on why observations of D17O have found lower values than produced by the model.**

We have added the following sentence to this paragraph:
"However, observations of $\Delta^{17}O$(nitrate) in autumn and winter in Beijing suggest much higher values (30.6±1.8‰) than was measured at Mt. Lulin (15 – 30‰ in winter).  A potential reason for the model overestimate of the observed values at Mt. Lulin could be qualitatively explained by transport of nitrate formed in the free troposphere to this high altitude location, where the high $\Delta^{17}O$(nitrate) producing pathways (R4-R8) should be negligible due to minimal aerosol surface area for heterogeneous chemistry."

**Page 12, line 7-14: This discussion is strange. The Savarino et al, 2007 work comes before the updated and much more certain (according to this manuscript) observation of D17O(O3). The error in _17O(O3*) has been reported to be 39+/-2 per mil, which seems to indicate that a tropospheric value as high as 41‰ could be possible. Further, Savarino's later work (Vicars and Savarino, 2014 cited in the manuscript! And comments of Savarino himself in presentations and in discussions on ACPD) in fact negates this conclusion suggesting that the photolysis and reformation of stratospheric ozone that enters the troposphere should reset the tropospheric ozone to local values (see discussion of this in Fibigier et al., JGR, 2016).**

It is important here to differentiate between transport of ozone versus nitrate from the stratosphere to the troposphere.  Indeed, transport of ozone from the stratosphere to the troposphere would not retain its stratospheric isotopic signature for very long (on the order of 3 hours as suggested by Michalski et al. [2014]).  Here we are referring to the transport of nitrate (not ozone) that was formed in the stratosphere and deposited to the surface. However, I do agree that the range in the observed values of $\Delta^{17}O(O_3{}^*)$ of 2‰ certainly allows for a value of 41‰ for nitrate formed within the troposphere assuming a $\Delta^{17}O(O_3{}^*)$ value at the upper end of the range and that all O-atoms of nitrate originate from ozone (*A* = 1 and all nitrate forms from R2 and/or R5).  Although this is not outside the realm of possibility for nitrate formed in the Antarctic troposphere during winter, it does seem unlikely that all nitrate in wintertime in Antarctica formed locally.  Since there are no known local source of $NO_x$ in the Antarctic winter, there must be a significant amount of nitrate formed at lower latitudes (where there is some sunlight and 41‰ would thus be unlikely) and transported to Antarctica.  We have added the following to the discussion:
"As previously noted in Savarino et al. (2007), the maximum observed $\Delta^{17}O$(nitrate) value (40.6‰) is not possible given our isotope assumption for the terminal oxygen atom of ozone

($\Delta^{17}O(O_3^*)$ = 39‰); however, it is theoretically possible given the 2‰ uncertainty in observed $\Delta^{17}O(O_3^*)$.  A value of $\Delta^{17}O(nitrate)$ = 41‰ is possible if $\Delta^{17}O(O_3^*)$ = 41‰ and all oxygen atoms of nitrate originate from ozone ($A$ = 1 and all nitrate forms from R2 and/or R5).  Although this may be possible for nitrate formed locally in the Antarctic winter due to little to no sunlight, lack of local $NO_x$ sources during Antarctic winter makes it unlikely that all nitrate observed in Antarctica forms locally. Long-range transport from lower latitudes and/or the stratosphere likely contributes to nitrate observed in Antarctica during winter (Lee et al., 2014)."

**Values near 40 per mil have also been observed in Greenland – by a different research group using different techniques – so it seems highly improbable that the values near 40 per mil simply cannot exist. Please update and reframe this discussion based upon more recent work and consider also acknowledging the observations in Greenland such as (Fibiger et al, JGR-Atmos., 121, 5010 5030, 2016) and references therein.**

A look at Fibiger et al. [2016] suggest values up to about 30‰, not 41‰.  There's a mention of 39‰ but this is an end-member extrapolation, not an observed value. Am I missing something? It would be nice to include this data set in the model-observation comparison; however, I cannot seem to find the actual data on the JGR web site or mention of where I can find it in the manuscript.

**Page 12 Lines 18-20: A more detailed comparison between modeled values and values observed in mid-latitudes should be made. The model results do not match well with results in Princeton despite the authors claims. In fact, the model matches better with La Jolla than with Princeton, so it is not clear why La Jolla is highlighted here. Point to figure comparisons more specifically here (ie Figure #'s).**

**The time series comparisons overall are not nearly as impressive as the global, bunched, comparison. What needs to be done in the community to get this more right? The time series speak to a lot of inconsistency in making local assumptions. For instance, there are important differences in model vs obs in the winter/spring of Princeton, Mt. Lulin, and La Jolla (and this likely speaks to the fact that local versus transported nitrate could be important); and then the fall values at Princeton, Mt. Lulin and Cape Verde are all not captured at all. More care should be taken and a more full discussion of the model/obs comparison should be done.**

Originally we focused on the largest discrepancies, i.e., the largest overestimates (Mt. Lulin) and the largest underestimates (polar winter).  We have added additional discussion of the discrepancies at all of the other locations shown in Figure 6 to this section.

"The model compares better to the mid-latitude locations close to pollution sources (La Jolla and Princeton), although the model overestimates wintertime $\Delta^{17}O(nitrate)$ in Princeton, NJ, USA by up to 6‰ and underestimates winter time $\Delta^{17}O(nitrate)$ in La Jolla, CA, USA by up to 4‰.  The model overestimate at Princeton during winter could be due to the fact that these are precipitation samples and not ambient aerosol samples, and thus may reflect nitrate formed at altitudes higher than we are sampling in the model.  The underestimate at La Jolla, CA could be

due to underestimates in reactive chlorine chemistry in the model, which would tend to increase $\Delta^{17}O$(nitrate) by increasing nitrate formation by the hydrolysis of halogen nitrates (R6) in this coastal location. The model underestimates the $\Delta^{17}O$(nitrate) observations at Cape Verde in late summer/early autumn by up to 6‰ (Savarino et al., 2013). Comparison with results from the steady-state model employed in Savarino et al. (2013) suggests that the low bias could be due to an underestimate of nitrate formation via $NO_3$ + DMS (R2). The steady-state model in Savarino et al. (2013) agreed with observations when R2 represented about one-third of total nitrate formation. The model results presented here have R2 representing about 15% of total nitrate formation in this season. An underestimate of the relative importance of R2 could result from a model underestimate of atmospheric DMS abundances."

We note that this added discussion of discrepancies at particular locations and times is speculative. A thorough comparison of the model with observations at individual locations would benefit from using the meteorology of the specific year of the observations (we ran only for the years 2015 and 2000) and a higher spatial resolution. The goal here is to present a comparison of all of the observations at once yielding a global perspective. This approach facilitates examination of isotopic assumptions in a way that comparisons at one location do not.

**Page 12-13: It would be useful to summarize here what impact the model uncertainties reported in the other works has on D17O(NO3-). Nowhere in the current work is the model compared to NOx or nitrate observations – only the isotopic composition of nitrate. So at least framing what uncertainties are important for consideration and the type of impact they would have on D17O(NO3-) seem important here.**

We have an entire section (section 4) following this section (section 3) devoted to discussion of model uncertainties utilizing several sensitivity studies. If you feel that something is missing from this section please specify.

It's true that we don't compare the model to observations of $NO_x$ and nitrate concentrations. Concentrations are dependent on many factors such as emissions, chemistry, transport and deposition, all of which have their own uncertainties. The advantage of $\Delta^{17}O$(nitrate) is that it is mainly sensitive to chemistry, and thus provides a metric to assess $NO_x$ chemistry in models in a way that concentration observations cannot.

**Page 13, lines 9-11 and line 27-28: I am not clear here why the uncertainty in the gamma(N2O5) is not considered here? E.g., the work by Bertram and Thornton (At-mos. Chem. Phys., 9, 8351–8363, 2009) and Tham et al. (Atmos. Chem. Phys., 18, 13155–13171, 2018) that suggest uptake coefficients range a lot from 0.001-0.035 and 0.005-0.039, respectively. How much uncertainty in D17O(NO3-) would changes in this parameter yield?**

The "cloud chemistry" model as presented here utilizes the Bertram and Thornton parameterization. As described in section 2, $\gamma_{N2O5}$ is calculated in the model as a function aerosol water content, chemical composition, and temperature and thus does vary over the

range you describe.  This would be better addressed in a paper comparing modeled and observed $\Delta^{17}O$(nitrate)  at a location and time period when the $N_2O_5$ pathway is dominant. Indeed, we are examining the importance of heterogeneous reactions in general for nitrate formation and $\Delta^{17}O$(nitrate) at a location (Beijing) where heterogeneous chemistry is likely very high.  This is a paper in preparation.

That said, we do examine the impact of the changing importance of the $N_2O_5$ pathway on $\Delta^{17}O$(nitrate) by comparing our "standard" and "cloud chemistry" simulations.  The cloud chemistry simulation results in an increase in $\Delta^{17}O$(nitrate) over the standard simulation due to the increase in the $N_2O_5$ pathway (compare Figure 5 and Figure S3) as a result of adding $N_2O_5$ hydrolysis in clouds.

**Page 16, Section 4.2: it should be made clear here why the "standard" simulation is used for this on not the cloud chemistry simulation, the latter is treated as if it is the state of the art through the rest of the manuscript.**

We decided to highlight the cloud chemistry simulation as it is the state of the science. However, this new cloud chemistry parametrization is very new, and is not yet included in any models (it is only now being implemented into the public version of GEOS-Chem).  Thus, all the sensitivity simulations were performed against the standard simulation of the model.  The conclusions drawn in the sensitivity simulations described in section 4.2 (hydrolysis of organic nitrates) and section 4.3 (photolysis of aerosol nitrate) should not change with the addition of cloud chemistry, as the cloud chemistry does not impact either of these reactions and the sensitivity simulations suggests that these uncertainties do not significantly impact the calculated $\Delta^{17}O$(nitrate) nor the conclusions.

**This section (and the previous) is really interesting. The authors should consider adding in figures of change in D17O(NO3-) based on the sensitivity studies. The emphasis is placed on gas phase chemistry changes in the figures, which is interesting, but since the paper is really about D17O(NO3-) it seems a missed opportunity to show some change in D17O. This is especially important in that the comparison with the time series observations (Figure 6) is underserved in the manuscript and makes the model seem much more uncertain. Regional digestion might speak to why they are such inconsistencies in seasonality at several stations in the mid-lats OR it might speak to how much difference in D17O is not captured by not having transported nitrate. Furthermore, future observational studies that compare with this work will be 1) better served, 2) this work will be more cited, and 3) this will advance the community forward in terms of our understanding of atmospheric chemistry based upon D17O (i.e. where we understand it and where we don't!).**

I made and considered adding figures showing the change in calculated annual-mean $\Delta^{17}O$(nitrate) for each of the sensitivity simulations described in section 4.  I decided not to show these figures because while the change in the *annual mean* $\Delta^{17}O$(nitrate)  is small, the change in a particular month or time of year can be significantly larger.  I was thus afraid that

showing the change in the annual mean $\Delta^{17}O$(nitrate) would imply that $\Delta^{17}O$(nitrate) is not very sensitive to nitrate production mechanisms, which is not the case.  I could show Figures 5 and 6 for each sensitivity simulation, which would not hide details that the annual mean hides.  I currently show these figures only for the "standard" and "cloud chemistry" simulations (Figures S3 and S4 compared to Figures 5 and 6).  This (difference between cloud chemistry and standard simulations) is the largest difference between sensitivity simulations (the difference between the other sensitivity simulations is smaller, as discussed in section 4).  If the editor wishes, I can add these additional figures (this would add 6 figures to the SI).  But again, the differences will be smaller than what is already shown.

**Page 16, lines 26-28: Globally, the D17O of nitrate remains relative unchanged from 2000 to 2015 emissions, but nitrate is not globally mixed. A more detailed regional analysis again would be really interesting here. For example, how does decreased NOx emissions impact the modeled D17O (and oxidation chemistry) and how does increased NOx emissions alter D17O in China? What are the implications for future observations?**

Please see the reply above.  This point is addressed in the text in section 4.2, which examines the impact of changing NOx emissions from 2000 to 2015 on nitrate formation pathways and $\Delta^{17}O$(nitrate).  The manuscript states:
"Relatively small changes (< 10%) in nitrate formation pathways yield small changes (< 2‰) in modeled annual-mean $\Delta^{17}O$(nitrate) between the year 2000 and 2015, differences in $\Delta^{17}O$(nitrate) over shorter time periods may be larger."

**Page 17, Conclusions: I again stress that the authors should make a full discussion of bulk versus transferred isotopic anomaly and the implications of previous assumptions. While it is compelling that the global model agrees better with the updated bulk and transferred value (and note that the transferred values reported by Vicars have an uncertainty of 2 per mil!), the global model still only explains 51% of the variance and the time series plots by location show important and significant disagreements. In other words, it is not a convincing assumption that because the global agreement is better with 25 per mil as the bulk that the observations are correct.**

We have rephrased our conclusions (and abstract and introduction) to avoid suggesting that the $\Delta^{17}O(O_3)$ value is now well known.  A thorough analysis of why the new observations of $\Delta^{17}O(O_3)$ may be incorrect is beyond the scope of this paper and would only be speculative. This issue is best addressed by a group other than the Savarino group repeating these nitrate-coated-filter measurements or utilizing another technique to measure $\Delta^{17}O(O_3)$ for comparison.

**As suggested above, having some discussion of regionality and figures of change in D17O(NO3) based upon the sensitivity studies would be useful, especially for observational work to compare with the model results and make progress on our understanding of key oxidation pathways.**

Please see previous responses to this point.

**This is minor, but it might be useful to acknowledge key assumptions in the work here and acknowledge what important steps forward are needed. Otherwise I imagine there will be a paper in another 10 years that will tell us "actually now we really know even better what we're doing" in atmospheric chemistry models.**

Hopefully our understanding of atmospheric chemistry will improve every 10 years!

**For instance, some key assumptions include: nitrate is locally produced; transported NO3- is not considered/treated nor is there any acknowledgement of how much of a difference this could make (see time series diffs!); NO emitted at night contains one-half its original O and one-half from local oxidant; the D17O(NO2) is calculated using 24-hour production rates [this is an improvement over earlier work but also means the production rates are out of sync with the nighttime versus daytime calculations]; model is compared with observations based on surface only.**

These assumptions and their impact on calculated $\Delta^{17}O$(nitrate) are all addressed explicitly in the manuscript. All of what is suggested in this comment is related to not transporting the isotopic tracers of NO, nitrate, and everything in between. We acknowledge up front in the manuscript that we don't transport the isotope tracers and discuss how this will lead to discrepancies, particularly at locations without local $NO_x$ sources. In order to quantify the effect at any particular location we would need to transport the tracers, which we do not do here due to the computational expense. However, as stated in the manuscript, the approach we use here will give the full range of calculated $\Delta^{17}O$(nitrate) values in the model which can be compared with observations. We think this is still quite useful for e.g., examining isotopic assumptions (for example, compare Figure 5 with Figure S5).

**Page 18, line 11: NO2+HO2 again? This not happening in the model correct?**

Please see previous reply to this point.

**Table 1: define A or refer directly to equation in text.**

Done.

**Figure 1: Consider that comparison (in the text) to the NO oxidation branching ratios of Alexander et al., (2009) would be interesting to allow for an understanding on how model updates have changed the modeled branching ratios with implications for D17O. Please make D's symbols in the fig caption.**

The text compares the global mean (80% versus 85% for NO + $O_3$). In Alexander et al. (2009), the rest (20%) is from NO + $HO_2/RO_2$. In the current version we also have NO + XO, which is small.

The symbols disappeared after uploading to ACPD.  I will resolve this issue with the final version.

**Figure 2: Adding an image of the D17O of NO2 would be interesting too.**

Good suggestion since other groups are trying to measure this.  I've added this figure to the SI (Figure S5).

**Figure 5: I'm not sure +/- 50% is really appropriate for this figure (also these are not identified in the figure caption). If the model were more than +/-25% off we could consider it completely not in the right world! It might be interesting to add the best fit line from Alexander et al. (2009) to compare with present study. References for the observational studies should be explicit in the figure or make a table and refer to that table.**

I'm not sure what you mean here by +/- 50%.  I think that showing a best fit line for data from another study that is not shown on the plot would be confusing to the reader.  References to the observations are in the text as stated in the figure caption.  Adding the reference list to the figure caption would make a long figure caption, but I'm happy to do this if the editor thinks it's appropriate.

**Figure 6: Again, it might be interesting to compare the time series with a what was predicted by the 2009 model. Reference for the observations need to be made. Delta should be a symbol. Why are there more than one observational point for DDU?**

The problem with what you suggest is that I cannot just use the data shown in the figures from the 2009 paper on this plot because different isotopic assumptions were made in the different studies, making the comparison misleading.  There were 2 year-long observation campaigns at DDU (Savarino et al. (2007) and Ishino et al. (2017) and I have shown each as separate data points.  Both of these studies are referenced in the text.

**Figure 7: Gammas should be symbols.**

Again, symbols disappeared somewhere between uploading and publication.  I'll fix all symbols in the final version.

**Figure 9: this is not particularly useful – it is exactly described in the text, could be moved to supplement.**

Agreed, it is now in the SI.

**Figure 10: Not sure "acidity-dependent yield" is how it is referred to in the main text? It is difficult to see these figures in this format. For Figs 7-11 I found myself wanting to understand how much change in D17O there would be associated with different regions.**

The acidity dependent yield is shown in Figure 9 (not 10), which was the previous comment and is now moved to the SI.

As far as the change in $\Delta^{17}O$(nitrate) for the sensitivity simulation, please see my response in previous comments.

**Supplement: Which simulation is used to produce Figure S1? Probably should make this explicit for all figures, or at least when it is NOT the cloud chemistry simulation. Is it possible to extend the color bar? It is difficult to digest since so much of the globe ends up close to 2 days.**

It is the "cloud chemistry" simulation. I've noted this in the figure caption. I chose to saturate the color bar at 2 days because extending the color bar makes it difficult to see the regions with lifetimes shorter than 1 day. It is the regions with the shorter lifetimes that are important for this part of the discussion, so I wanted to make sure they are clear.

**Figure S2-S6: suggest including a more complete caption stating that this is . . . then same as. . .or for comparison with Figure...**

These have all been changed except for Figure S5. I don't want the different isotopic assumption made in this figure to get lost in a long figure caption.

**Figure S5 caption is incorrect?**

This has been fixed. It is the same as Figure 5, not S1.

---

## Author Comment (AC4) · 20 Sep 2019

**Reviewer comments in bold**, author responses in plain text.

**The discussion on A values on page 8 has a serious flaw, namely it ignores the rapid isotopic exchange between NO and NO2 (Sharma) and N2O5. This means that the _17O = 0 NO emitted at night does not have to be oxidized into NO2 to dilute NO2_17O value, but can simply exchange with existing NO2. Likewise, nighttime equilibrium NO3+NO2< ! N2O5 would ultimately incorporate additional ozone into NO2.**
**In other words there is a serious limitation to the counting oxidations and ignoring the exchanges during the nighttime. It probable that that at night isotope exchange equilibrium results in _17O of NO = NO3 = NO2 This in turn would impact HONO _17O and NO2 "cloud chemistry" at night and HNO3 production early morning when O3 levels are low due to nighttime titration.**

Thank you for this point.  I do think it's important that we discuss this isotopic exchange in the manuscript; however, it won't impact our isotopic assumptions. Isotopic exchange between NO and $NO_2$ may increase $\Delta^{17}O(NO)$, but it will decrease $\Delta^{17}O(NO_2)$ by the same amount (isotopic mass balance).  Similarly, isotopic exchange between $NO_2$ and $NO_3$ (via the $N_2O_5$ intermediate) may increase $\Delta^{17}O(NO_2)$, but it will decrease $\Delta^{17}O(NO_3)$ by the same amount.  So our assumed value of $N_2O_5$ won't change, and thus the calculated value of $\Delta^{17}O(nitrate)$ from $N_2O_5$ hydrolysis (R4) won't change.  Remember, this is a global model so we aren't keeping track of individual molecules but are making assumptions about the bulk isotopic composition within a grid box.  Of course, atmospheric measurements also represent a bulk quantity.  We have updated our discussion of $\Delta^{17}O(NO_x)$ during the daytime versus the nighttime in the introduction and methods sections and added appropriate references.

**The other serious limitation is the treatment of the ozone _17O value. It is well known that _17O and _18O in ozone is a strong function of temperature and pressure. The choice of Vicars (Over cryogenic collection studies) because of the apparent constant _17O values is because these were all surface measurements at effectively the same pressure and a narrow temperature range. It is unlikely O3 being recycled above the boundary layer will have a 26 per mil _17O. How much nitrate is formed in the mixed layer versus free troposphere?**

I don't see how this matters. You say in your 2014 paper in ACP that ozone transported from the stratosphere into the troposphere won't retain its stratospheric isotopic signature because the isotopic lifetime of ozone is short in the troposphere due to rapid ozone photolysis and reformation.  Why would this be any different for ozone transported from the free troposphere to the boundary layer?

**Also the authors have chosen to ignore our Atmos. Chem. Phys., 14, 4935–4953, 2014 paper where we showed the pressure and temperature dependence in NO2 _17O values in**

**equilibrium with O3 as a function of temperature and pressure that demonstrates this effect. A lot hinges on the validity of "Recently, much more extensive observations of ïA¸Dˇ 17O(O3) using a new technique (Vicars et al., 2012) show ïA¸Dˇ 17O(O3) = 26 ± 1‰ around the globe (Vicars et al., 2012;Ishino et al., 2017b;Vicars and Savarino, 2014), and suggest that previous modeling studies are biased low in ïA¸Dˇ 17O(nitrate) (e.g., Alexander et al. (2009)), which would occur if the model underestimated the relative role of ozone in NOx chemistry."**

**These are nearly all clean marine boundary layer measurements and simply ignoring the Johnston and Krankowsky cryogenic collection is polluted urban environments seems to be cherry picking the data. Likewise our experimental NO2 _17O values match well with that predicted by the T and P dependence of O3 formation experiments (i.e Thiemens, Mauersberger group). This is not the first paper to ignore these unpleasant contradictions. It seems no ones wants to acknowledge that something we do not understand is going on with either tropospheric O3 _17O dynamics or their measurements.**

I initially neglected any discussion of the potential uncertainty in the $\Delta^{17}O(O_3)$ observations using the nitrite coated filter technique as it has yet to be specifically shown that there are problems with this.  However, I see your point that this also has yet to be replicated by other groups.  Another reviewer also had this same issue.  I have now have modified some wording and added additional discussion so as not to place too much certainty in this value of $\Delta^{17}O(O_3)$ = 26‰.  Wording changes are in the abstract, introduction, and conclusions.

Your 2014 paper should have been cited in the original manuscript and we have added this citation in several locations in the revised manuscript.

**Also they might include Wang et al. for some additional south American data**
**https://www.sciencedirect.com/science/article/pii/S0016703714001811?via%3Dihub**

Thanks for this suggestion.  It would be great to include this in Figure 5; however, the data is not provided in the paper or in the supplement (it is only plotted).  Due to the low spatial resolution of the model, including this data would add one data point to Figure 5.  The $\Delta^{17}O$(nitrate) observations from this paper represent a 3.5 year mean value.

---

## Author Response (AR2)

We thank the editor and reviewers for their thoughtful comments. We have responded to each
comment and made appropriate changes to the manuscript. **Reviewer comments are in bold,** author
responses are in plain text. A tracked-changes version of the manuscript and the SI is appended below
our responses.

**Editor comments:**

**Introduction (p. 5/l. 9) and conclusions (19/12):**
**Please add a caveat that previous modelling efforts have made different assumptions about the**
**preferential transfer of central and terminal O atoms to NO2 and NO3, and the 17O enrichment of**
**different ozone isotopomers. This is still not clear enough.**

This has been added to the last paragraph of the Introduction. It now reads:

"Previous modeling studies showed good agreement with observations of $\Delta^{17}O$(nitrate) when assuming
that the bulk oxygen isotopic composition of ozone ($\Delta^{17}O(O_3)$) is equal to 35‰ (Alexander et al.,
2009;Michalski et al., 2003), but varied in their assumption on terminal oxygen atom versus statistical
isotopic transfer from $O_3$ to the reactant (NO and $NO_2$). This is an important distinction because it is
now known that the $^{17}O$ enrichment in $O_3$ is contained entirely in its terminal oxygen atoms, and it is the
terminal oxygen atom that is transferred from $O_3$ (Vicars et al., 2012;Berhanu et al., 2012;Bhattacharya
et al., 2008;Savarino et al., 2008;Michalski and Bhattacharya, 2009;Bhattacharya et al., 2014), so that
the $\Delta^{17}O$ value of the oxygen atom transferred from ozone to the product is 50% larger than the bulk
$\Delta^{17}O(O_3)$ value."

Some of the wording above was previously at the end of the Methods section, and has been removed to
avoid unnecessary repetition.
**5/2: Remove tilde sign and adjust interval to encompass full range of observations (6 to 54 ‰ based**
**on Krankowsky et al. 1995; 19 to 41 ‰ based on Johnston & Thiemens 1997).**
Done.
**5/4: Likewise, the range shown here is too narrow. It's 30 to 46 ‰ for Morton et al. (1990). Please**
**also add "et al." to the reference.**

Done.
**17/26 & 20/1: Replace tilde sign by actual range value with uncertainty. All measurement results**
**should be rounded according to their uncertainty and stated with an estimate of their measurement**
**uncertainty. Approximation symbols should therefore not be used (unless you are approximating a**
**mathematically exact number, e.g. π ≈ 3.14). In any case, the correct approximation symbol has two**
**wavy lines (≈). It is not the tilde sign (~), a symbol which has perhaps made it into the literature due to**
**limitations of early typewriters.**

Thank you for this point. I have included the exact range. As part of this I found a typo, what said
"increases" should have said "decreases".

**Figure S1: More than half of the plot appears with the color corresponding to the colorbar maximum.**
**Please include a variant of the plot with an increased maximum value so that variations in τ >= 2 d can**
**be distinguished, or perhaps add contour lines for values higher than 2 days.**

I have remade this plot on the log scale and included the full range of calculated values.

**Figure S3: Please explain the meaning of the dashed lines in the figure caption.**

I have added the following to the Figure 5 and Figure S3 captions:

"The y=x (solid line) and y = 2x and y = 0.5x (dashed) are shown."

**Figure S6: The caption should refer to Fig. S3.**

Thanks for catching this. It has been fixed.

**Anonymous Referee #2:**

**The authors have certainly improved the manuscript in response to the reviewer's comments.**
**Submission of the revised manuscript and continuing onto publication in ACP is warranted. There are**
**few areas that the authors should revisit and consider further revision based upon the original**
**reviewers' comments:**

**(1) The authors added a qualitative explanation for the lack of agreement with observations in Mt.**
**Lulin as lack of heterogeneous chemistry "due to minimal aerosol surface area." However, this**
**statement contrasts with the conclusions drawn in the Guha et al. observational study, so the**
**response by the authors needs to be refined to better explain this interpretation (do they mean that**
**the model predicted aerosol surface area is too lacking to have heterogeneous chemistry?).**

Indeed there does seem to be a discrepancy between the interpretation of the observations at Mt. Lulin
in Guha et al. with both subsequent observations in Beijing and in the model.  I point out the former by
stating that although the authors of the Mt. Lulin paper state that nitrate is transported to Mt. Lulin
from polluted regions, that this is not consistent with the observations in Beijing, which show much
higher $\Delta^{17}O$(nitrate) values than what was measured at Mt. Lulin.  If transport from polluted regions was
the reason for the model-observation discrepancy, one might expect that the model would
underestimate the observations, and the opposite occurs. Thus I'm suggesting that the reason for the
model-observation discrepancy is that this location receives transport from the free troposphere, where
$NO_2$+OH dominates nitrate formation. To make this more clear, I have added the following sentence to
this paragraph:

"Low $\Delta^{17}O$(nitrate) values from nitrate formed at higher altitudes and transported to Mt. Lulin would not
be accounted for in the model since the isotopes are not transported."

**(2) The authors make the excuse in considering a comment about the Wang et al., GCA, 2014 paper**
**and the Fibiger et al., 2016 paper that the data is "not available". It has long been the practice to**

**contact corresponding authors for data if it is not available in the manuscript. And I found that the**
**Fibiger et al., 2016 reference actually states the following: Data from this paper are available at**
**ACADIS. Data sets https://www.aoncadis. org/project/collaborative_research_the_**
**impact_of_bromine_chemistry_on_the_ isotopic_composition_of_nitrate_at_**
**summit_greenland.html.**

**Looking at this website it appears to include the isotope data from both Fibiger et al 2016 and Fibiger**
**et al 2013 (the 2013 one reports the D17O data). The D17O data from the Fibiger et al., 2013**
**(Geophysical Research Letters, VOL. 40, 3484–3489, doi:10.1002/grl.50659, 2013) should be**
**considered in the current study and does include values that look to be close to 39 per mil (or at least**
**definitely >>30 per mil!). The authors should revisit this and consider the implications for their**
**response in the manuscript. Also consider contacting F. Wang or G. Michalski for the data from Wang**
**et al. so this can be included as well.**

I have contacted the authors of these papers and obtained the data.  I have included the data from
Fibiger et al. [2013] in Figures 5 and 6 (and in the related figures in the supplement) and the Wang et al.
[2014] data in Figure 5.  I include only the concentration weighted, monthly mean measurements from
Summit in June of 2010 and 2011. Although there were also measurements in May, it was only for the
second half of May. Since May is in the shoulder season, there may be a significant difference between
early and late May, and I have only output monthly means from the model.  This adds two data points to
Figures 5 and 6.  The error bars in Figure 6 for the Fibiger et al. data reflect the standard deviation of the
measurements, and this is stated in the figure caption.  The Wang et al paper adds one data point to
Figure 5. Although there were measurements at 9 different locations, all 9 locations are in the same
model grid box.  I calculated the concentration weighted monthly mean from observations at all 9
locations, and compared with the mean $\Delta^{17}O$(nitrate) from the model from July – December, which is
when the measurements occurred.  In sum, these data sets add 3 additional data points to Figure 5, and
together do not change the statistics.

**(3) The statement added by the authors that "Although lack of transport of the isotope tracers hinders**
**direct comparison of the model with observations at any particular location" contrasts with the fact**
**that they make direct comparison with a range of time series in Figure 5. So maybe restate this that**
**the lack of transport adds uncertainty to direct comparisons – but you do make direct comparisons in**
**space and in time!**

Thanks for this suggestion. This has been changed to the following:

"Although lack of transport of the isotope tracers adds uncertainty to direct comparison of the model
with observations at any particular location, …"

**(4) The phrasing of "the influence of clouds on nitrate formation" does not really make sense. This**
**should be rephrased to account for the fact that precipitation will represent a column average of**
**nitrate (i.e. long-range transported nitrate, nitrate formed in clouds, and nitrate formed near the**

**surface). The point that the meteorology tends to have clouds near 1 km means that the model**
**sampling is robust for comparison on this point, but the impact of clouds on nitrate formation does**
**not seem to be the point here.**

I am referring to the influence of clouds on the *chemistry* of nitrate formation here, not on the influence
of wet deposition on nitrate abundance. This is because the model now includes nitrate formation
chemistry in cloud droplets. For clarity, this has been restated as follows:

"However, since cloud water peaks on average near 1 km altitude in the MERRA2 meteorology used to
drive GEOS-Chem, our model sampling strategy should capture the majority of the influence of clouds
on *the chemistry of* nitrate formation."
**(5) It is not clear whether the authors added a clear reasoning for the cloud chemistry simulation**
**versus the standard simulation to the manuscript. (Response to comment marked "Page 16, Section**
**4.2"). The manuscript needs to be clear about how, when and where the results from different**
**simulations are being used and why.**

We state in the paper that we focus on the "cloud chemistry" simulation because we consider it the
state of the science.  All other simulations are presented in Section 4, as stated here in the manuscript.
We have added a justification for why we conduct sensitivity simulations relative to the "standard"
model, as shown below:

"Additional model sensitivity studies are also performed and examined relative to the "standard" model
simulation, which represents a more common representation of nitrate chemistry in atmospheric
chemistry models."

**(6) Regarding the comments on understanding D17O(NO3-) more regionally (e.g., showing how**
**D17O(NO3-) changes regionally based upon the sensitivity studies). Perhaps another way to consider**
**this is to add a figure to the SI that shows the results of the different simulations for the times series**
**comparison with observations (ie Figure 6). This would give much more quantitative information for**
**researchers conducting observations and give much more information about how sensitive the D17O**
**is in different regions where these processes are more/most important in different seasons. This**
**would only add 1 figure to the SI (i.e. Figure 6 with different color lines representing a few different**
**sensitivity simulations?).**

I have replaced Figure S4. The old Figure S4 showed the results from the "standard" simulation. The new
Figure S4 shows results from all of the simulations (total of 7). A figure with different colors for each
simulation was hard to read because of the large number of simulations.  Instead I show the "cloud
chemistry" simulation again as points, but with error bars reflecting the full range from all sensitivity
studies.  In the main text (Figure 6), the error bars are different, and instead reflect the estimated
impact of assuming isotopic equilibration in Mt. Lulin, which is near populated regions in China where
nighttime nitrate formation is relatively fast.

**(7) The dashed lines in Figure 5 appear to represent +/- 50%. These should be defined in the figure**

**caption and the authors should consider whether it would make more sense to include dashed lines at +/- 25%.**

This is now explicit in the captions of Figure 5 and Figure S3.

**Greg Michalski:**

**The authors have substantially improved their manuscript. However I believe they need more throughly and directly address two issues raised by several of the reviewers.**

**1) The troublesome of value of the O3 D17O value as some fixed value. Using Vicars et al. data does not address the T and P effect demonstrated by numerous lab experiments. The argument that stratospheric O3 "resets" avoids the issue. Any NO oxidation or NO3- production above the mixed layer will likely have a different D17O because the O3 D17O in those layers will be a function of T and P and not fixed at 25 permil. The authors seem to argue that using 25 best "fits the data". This seems a circular argument. One could also argue that the experimental O3 D17O are correct and the pathways are actually wrong. There should be a measure of NO3- production in each model layer...How important is NO3- production at say 5 km and what might the O3 D17O be t this T and P? It would be difficult to hash all this out in the current paper but my fear is that there is a mantra of "its 25 permil always and everywhere" is being repeated by a host of recent papers at the expense of numerous other studies that say otherwise. This makes it increasing difficult to challenge. There should be a least one paragraph that there is somethings we don't understand about O3 D17O and a critical assessment of these conflicting estimates.**

You are correct that the $\Delta^{17}O(O_3)$ observations from Vicars et al. are at the surface, and thus may not represent the value of $\Delta^{17}O(O_3)$ in the free troposphere. Fortunately for this model-observation comparison, the $\Delta^{17}O(nitrate)$ observations are also at the surface. I've added some additional discussion on this topic to the last paragraph of the introduction. The end of this last paragraph now reads:

"Note that laboratory studies show that the magnitude of $\Delta^{17}O(O_3)$ is dependent on temperature and pressure (Heidenreich and Thiemens, 1986;Thiemens, 1990;Morton et al., 1990). The observations of $\Delta^{17}O(O_3)$ by Vicars et al. (2012, 2013) were at the surface over a large temperature range, but may not reflect the value of $\Delta^{17}O(O_3)$ at higher altitudes. However, with the exception of lightning, whose emissions are presently several times smaller than $NO_x$ emissions from anthropogenic and biomass burning sources (Murray, 2016), $NO_x$ sources emit at the surface. With a $NO_x$ lifetime relative to its conversion to nitrate on the order of one day (Levy et al., 1999), most nitrate formation also occurs near the surface. Here, we examine the relative contribution of each nitrate formation pathway in a global chemical transport model and compare the model with surface observations of $\Delta^{17}O(nitrate)$ from around the world."

**2) The role of NO emissions at night is still not satisfactory addressed. Morin et al.s model did not**

**include emissions, thus their conclusions about 5% are not valid. In most of the domain of a global**
**model the nighttime emissions are comparable to daytime. Only urban areas with vehicles is there a**
**significant difference between daytime and night emissions. Thus NO emitted at night retains its**
**source O until sunrise scrambling. How much of this oxidized at night to NO2 to exchange or form**
**NO3-? Clearly this would have a major impact in high latitudes in the winter. Are we to be convinced**
**the NO emitted in Alaska in Jan. is photochemically equilibrated with O3 within 5 %? Seem**
**implausible. I do not expect the authors to redo their model, but there should be another full**
**paragraph is the discussion of the limits of the equilibration assumption.**

I agree that the results of Morin et al are not valid since they did not emit NO at night. I've deleted the sentence referencing this paper.

To estimate the error due to the assumption of isotopic equilibration of $NO_x$ in the model, we calculate the lifetime of $NO_x$ against oxidation to nitrate from the chemical pathways that only occur at night. This is plotted in Figure S1 (which has been revised to show the full range of calculated values).  The shorter the $NO_x$ lifetime against nighttime oxidation, the more likely it is that NO emitted at night will be oxidized to nitrate before sunrise.  Figure S1 shows that the shortest lifetime against nighttime oxidation is 0.4 days and that lifetimes less than one day occur in only very few locations. Over the majority of the globe, the lifetime of $NO_x$ against oxidation at night is > 1 day, suggesting that the majority of NO emitted at night will survive until sunrise prior to oxidation to nitrate.

We investigate the uncertainty in the assumption of $NO_x$ isotopic equilibration by assuming that half of total nitrate measured forms at night from NO that was emitted during that same night (i.e., $NO_x$ is not isotopically equilibrated during the daytime before being oxidized to nitrate).  This effectively assumes that all nitrate emitted at night is oxidized at night prior to sunrise, which is very likely an overestimate of the true bias.  We make this calculation for Mt. Lulin, because it is in a region (China) with $NO_x$ lifetimes against nighttime oxidation that are less than one day.  This uncertainty is represented as error bars for this location in Figure 6, and as you can see cannot account for the model-observation discrepancy.  If this assumption were an issue in the model, one would expect that the model would overestimate $\Delta^{17}O$(nitrate) in such regions; however, the opposite is the case for Beijing, where the model underestimates the observations (as shown in Figure 5 and discussed in the text).

Certainly if $NO_x$ is emitted at a high enough latitude that experiences 24-hours of darkness during winter, there will be no photochemical isotopic equilibration.  However, it is also likely that any nitrate measured at that location will have formed at lower latitudes and transported to higher latitudes, as $NO_x$ emissions in polar regions have very low (if any) local $NO_x$ emissions.

For your Alaska example, it will depend on location. Alaska is a big state, and the most northern parts may experience 24-hours of darkness. Fairbanks, for example, does not fall into this category, as it has over 3 hours of sunlight on the winter solstice.  It would certainly be an interesting case study though. Since the winter days are short and air pollution can be quite high, one might expect this to be a location that would experience nighttime oxidation fast enough (long nights with high aerosol surface area) that a significant fraction of NO is both emitted and oxidized at night prior to sunrise. I know that the Savarino group is measuring both $\Delta^{17}O(NO_x)$ and $\Delta^{17}O$(nitrate) at this location, and I look forward to seeing their results as it will be a nice observational constraint on the magnitude of the bias in the model when assuming photochemical equilibrium.

[revised manuscript text omitted]

Comparison of monthly-mean modeled ("standard") and observed $\Delta^{17}$O(nitrate) at locations where there are enough observations to calculate a monthly mean. References for the observations are in the text. The error bars represent different assumptions for calculated modeled *A* values for nighttime reactions as described in the text. Error bars for Beijing and Mt. Lulin reflect the range of possible modeled *A* values for nighttime reactions as described in the text. The y=x (solid line) and y = 2x and y = 0.5x (dashed) are shown.

[Figure]

**Figure S4.** Comparison of monthly-mean modeled and observed $\Delta^{17}$O(nitrate). Model points are from
the "cloud chemistry" simulation, while the modeled error bars reflect the full range of calculated values
from all sensitivity simulations.  Error bars for the observations reflect the analytical uncertainty in the measurements, except for two data points in June for Summit which reflect the standard deviation of $\Delta^{17}O$(nitrate) from multiple measurements during that month.

[Figure]

Figure S5. Modeled, annual-mean $\Delta^{17}O$(NO2) below 1 km altitude for the "cloud chemistry" model.

[Figure]

**Figure S6.** Same as Figure S3 but assuming $\Delta^{17}O(O_3)$ = 35‰.

[Figure]

**Figure S7.** Same as Figure 3 but for the "standard" simulation.

[Figure]

Figure S8. Calculated yield of HONO from the heterogeneous reaction of $NO_2$ on aerosol surfaces as a function of pH.

[Figure]

Figure S9. Calculated surface aerosol pH in the model in each season.

[Figure]

**Figure S10.** Modeled change in anthropogenic NO emissions (Gg N yr$^{-1}$) from the year 2000 to the year
2015 (2015 − 2000).